



# Sap flow and leaf gas exchange response to drought and heatwave in urban green spaces in a Nordic city

Joyson Ahongshangbam[1,4], Liisa Kulmala[2,3], Jesse Soininen[1], Yasmin Frühauf[1], Esko Karvinen[2], Yann Salmon[1,3], Anna Lintunen[1,3], Anni Karvonen[1], and Leena Järvi[1,4]

[1]Institute for Atmospheric and Earth System Research (INAR)/Physics, Faculty of Science, University of Helsinki, Helsinki, Finland
[2]Finnish Meteorological Institute, Helsinki, Finland
[3]Institute for Atmospheric and Earth System Research/Forest Sciences, Viikki Plant Science Centre (ViPS), Faculty of Agriculture and Forestry, University of Helsinki, Helsinki, Finland
[4]Helsinki Institute of Sustainability Science, University of Helsinki, Finland

**Correspondence:** Joyson Ahongshangbam (joyson.ahongshangbam@helsinki.fi)

**Abstract.** Urban vegetation plays an important role in offsetting urban $CO_2$ emissions and mitigating heat through tree transpiration and shading. With frequent heatwave events and the accompanying drought, the functioning of urban trees is severely affected in terms of photosynthesis and transpiration rate. The detailed response is however still unknown despite tree functioning having crucial effects on the ecosystem services they provide. We conducted sap flux density ($J_s$) and leaf gas exchange

measurements of trees (*Tilia cordata*, *Tilia × europaea*, *Betula pendula*, *Malus spp.*) located at four types of urban green areas (Park, Street, Forest, Orchard) in Helsinki, Finland, over two contrasting summers 2020 and 2021. Summer 2021 had a strong heatwave and drought, whereas summer 2020 was more typical for Helsinki. In this study, our aim was to understand the responses of urban tree transpiration and leaf gas exchange to heatwave and drought and examine the main environmental drivers controlling the transpiration rate during these periods in urban green areas. We observed varying responses of tree water

use during the heatwave period at the four urban sites. $J_s$ was found to be 35-67% higher during the heatwave as compared to the non-heatwave period at the Park, Forest, and Orchard sites but no significant difference was found at the Street site. Our results showed that $J_s$ was higher (31-63%) at all sites during drought as compared to non-dry periods. The higher $J_s$ during the heatwave and dry periods were mainly driven by the high atmospheric demand for evapotranspiration represented by the vapor pressure deficit (VPD), suggesting that the trees were not experiencing severe enough heat or drought stress

that stomatal control would have decreased transpiration. Accordingly, photosynthetic potential ($A_{max}$), stomatal conductance ($g_s$), and transpiration ($E$) at the leaf level did not change at the four sites during heatwave and drought periods. Only $g_s$ was significantly reduced during the drought period at the Park site. VPD explained 55-69% variations in the daily mean $J_s$ during heatwave and drought periods at all sites except at the Forest site where the saturation of $J_s$ at high VPD was evident due to low soil water availability. The heat and drought conditions were untypically harsh for the region but not excessive enough to

restrict stomatal control and the increased transpiration indicating that ecosystem services such as cooling was not at risk.

Keywords: climate extreme, drought, sap flux, transpiration, urban trees



## 1 Introduction

Urbanization is increasing rapidly and transforming the natural environment, land cover, and ecological functions. Urbanization enhances $CO_2$ emissions and the local urban heat island (UHI) effect, leading to harsher conditions in cities and a decrease in human thermal comfort when compared to more natural surroundings (Oke et al., 1989; Roth et al., 1989). These challenges highlight the need for adequate urban planning in the long-term development of urban areas. Urban green areas play a vital role in compensating urban $CO_2$ emissions and mitigating the UHI effect as they have the potential for carbon sequestration and storage, and regulation of water and surface energy balance (Lindén et al., 2016; Bowler et al., 2010). Urban trees also provide other ecosystem services such as cooling effect through shading, pollutant infiltration, aesthetics and recreation, buffer for noise and wind, and soil conservation (Bussotti et al., 2014; Berland et al., 2017). Several studies have highlighted the $CO_2$ sequestration potential and annual carbon storage in urban vegetation (Nowak and Crane, 2002; Davies et al., 2011; Muñoz-Vallés et al., 2013; Nowak et al., 2013), and addressed urban green areas as a way of mitigating the global GHG emissions in cities (Dhakal, 2010; Paloheimo and Salmi, 2013; Pataki et al., 2021). The role of urban trees in mitigating UHI through shading by tree canopies and cooling effect by transpiration has been reported in many urban studies (Rahman et al., 2019; Pataki et al., 2011). Urban trees are subjected to human disturbances and climate change impacts including extreme weather such as heatwaves and drought. All these affect the potential of trees to mitigate and adapt to climate change, and thus it is important to understand the response and functions of urban trees during extreme climate events.

In general, environmental conditions for urban trees are often more extreme than those in a natural forest stand, having e.g. higher air temperature, lower air humidity, and more limited soil water and nutrient availability (Nielsen et al., 2007). All these differences in local microclimate, growing conditions, species type, disturbances and management activities affected the functioning of urban trees. The rise in temperature during summer heatwaves affects leaf temperature potentially leading to leaf damages (Kunert et al., 2022; Atkin and Tjoelker, 2003; Ghannoum and Way, 2011), and vapor-pressure deficit (VPD) affecting transpiration and photosynthesis through stomatal control (Lloyd and Farquhar, 2008). The response of urban trees to climate change has also been found to differ from that of forest trees and moreover, urban trees have been found to be more susceptible to multiple stresses enhanced by climate change (Bussotti et al., 2014). Droughts have occurred more frequently in recent times causing severe symptoms for trees even in areas that are considered to be rather moist high latitude areas (Hartmann et al., 2022). Drought usually co-occurs with high air temperature during summer severely affecting the functioning of urban trees and potentially resulting in leaf damage, reduced carbon assimilation and transpiration through lowered stomatal conductance (Bussotti et al., 2014; Winbourne et al., 2020).

However, the effect of an urban environment on the tree's response to stress such as drought and heatwaves are not known, particularly in the boreal urban environment. A few studies have investigated the potential cooling effect of urban trees during heat and drought (Gillner et al., 2015b), and the impact of drought on urban tree functions (Rötzer et al., 2021). The urban tree functions were found to be affected by species type, growing conditions, local climate and water availability, particularly during heatwave and drought periods. Nonetheless, urban tree responses to extreme heat and drought are poorly addressed and, with complex urban stressors and heterogeneity, need further study. Especially information on the impact of heat and drought





on urban trees in high-latitude cities is lacking. Global warming is more prominent at high latitudes making urban ecosystems in high latitudes particularly vulnerable. Thus, it is important to understand the responses of trees to heat and drought in various urban environments in the boreal region.

In this study, we measured the functions of trees, particularly the transpiration and leaf gas exchanges, during heatwave and drought periods in a boreal urban environment in Helsinki, Finland. In particular, our focus was on the variability of tree water use patterns in different urban green areas, and on the environmental controls of tree water use. We addressed the following research questions:

1. How does heatwave affect transpiration and leaf gas exchange rates of urban trees?
2. How does drought affect transpiration and leaf gas exchange rates of urban trees?
3. How do environmental drivers affect transpiration rates during heatwave and drought periods in urban green areas?

We hypothesize that H1) heatwave increases transpiration due to increased VPD which might cause a decrease in the photosynthetic potential due to lower stomatal conductance, H2) drought decreases both transpiration and the photosynthetic potential, and H3) the role of VPD as a driver decreases during heatwave and drought periods, again due to lower stomatal conductance. To answer these questions and test the hypotheses, we conducted continuous sap flux and manual leaf gas exchange measurements at four urban green areas (Park, Street, Forest and Orchard) in Helsinki during the summers of 2020 and 2021. The summer of 2021 was hot and dry as the mean air temperature in July 2021 was 21.6°C, being 21% higher than in July 2020 (16.7°C) and 19% higher than the average mean temperature in July (18.1 °C) during a climatic reference period (1991 to 2020). The total precipitation for the months of June and July 2021 (86 mm) was 51% lower than during June and July in 2020 (177 mm) and 27% lower than the average total precipitation in June and July during the climatic reference period (117 mm).

## 2 Methods

### 2.1 Sites description

The study was conducted in the vicinity of the University of Helsinki Kumpula campus, located 4 km northeast of the Helsinki city centre. The Kumpula area is characterized by heterogeneous land-use cover (Figure 1a), particularly by diverse urban vegetation. Within the study area, four sites were selected to be studied: a park with sparse trees ('Park'), a roadside plantation ('Street'), an urban forest ('Forest'), and an apple orchard ('Orchard'). These sites are located close to a micrometeorological eddy covariance station (FI-Kmp, 60°12'11.3"N 24°57'40.4"E) which is also an Associated Ecosystem Station of ICOS (Integrated Carbon Observation system), and part of the SMEAR (Station for Measuring Ecosystem Atmosphere Relations) III station (Vesala et al., 2008; Järvi et al., 2009). Overall, Helsinki is a humid continental region (according to Köppen climate classification), with annual precipitation of 652 mm yr$^{-1}$ and annual mean temperature of 6.5 °C during the 30-year climatic reference period 1991-2020 (FMI, 2021).



The urban park (Park; Figure 1b) is located in the Kumpula botanical garden, South-west of the FI-Kmp measurement tower. The site is characterized by a mixture of *Tilia species* trees, and a ground layer of short vegetation comprised mainly of lawn species, clovers (*Trifolium repens*) and mosses. The ground vegetation was mowed using an automatic mowing device leaving the clippings on-site and irrigation was activated on dry and warm days within the wider park area. However, the mowing and irrigation were restricted in-between the study trees (0.25 ha area) during the measurement period but the tree roots reached the irrigated area.

The roadside plantation (Street; Figure 1 c) is located on the Hermannin rantatie road, 0.8 km east of the FI-Kmp measurement tower. It consists of a row of *Tilia × europaea*, which is a hybrid of *T. cordata* and *T. platyphyllos*. The species is the most commonly planted urban tree in Helsinki and Nordic countries in general, comprising 44% of Helsinki's Street trees (Sjöman et al., 2012). The street trees grow over a soil patch spreading 60 m long and 2.7 m wide, with an average tree spacing of 8.2 m. Normally, the trees are regularly trimmed and maintained by the city gardening company but they were non-managed during our study.

Urban forest (Forest; Figure 1d) is a small forest patch (25x30 $m^2$) located between the FI-Kmp measurement tower in the north and the Kumpula garden in the south. This site is dominated by mature *Betula pendula* trees. This site was the only non-managed study site and was regenerated naturally. Other deciduous trees such as *Betula pubescens, Alnus glutinosa, Acer platanoids*, and *Ulmus glabra* were also found in this urban forest as a species mixture. The ground layer was sparse consisting mainly of *Aegopodium podagrariaa* and bare spots.

The apple orchard (Orchard; Figure 1e) is located in the Kumpula school garden 0.9 km west of the FI-Kmp measurement tower. The site is characterized by scattered apple trees (20 trees per 30 m x 30 m area) planted over a managed lawn. There was no irrigation. The lawn in our measurement area was manually mowed a few times during the summer.

More detailed descriptions of all four sites are given in Table 1 and of soil properties in Appendix A. All sites were equipped with continuous measurements of sap flux and meteorological variables from June 2020 to September 2021, except the Orchard site, where data was recorded only from June-September 2021. The continuous measurements were accompanied by manual measurements of leaf gas exchange during the summer months (June-August).





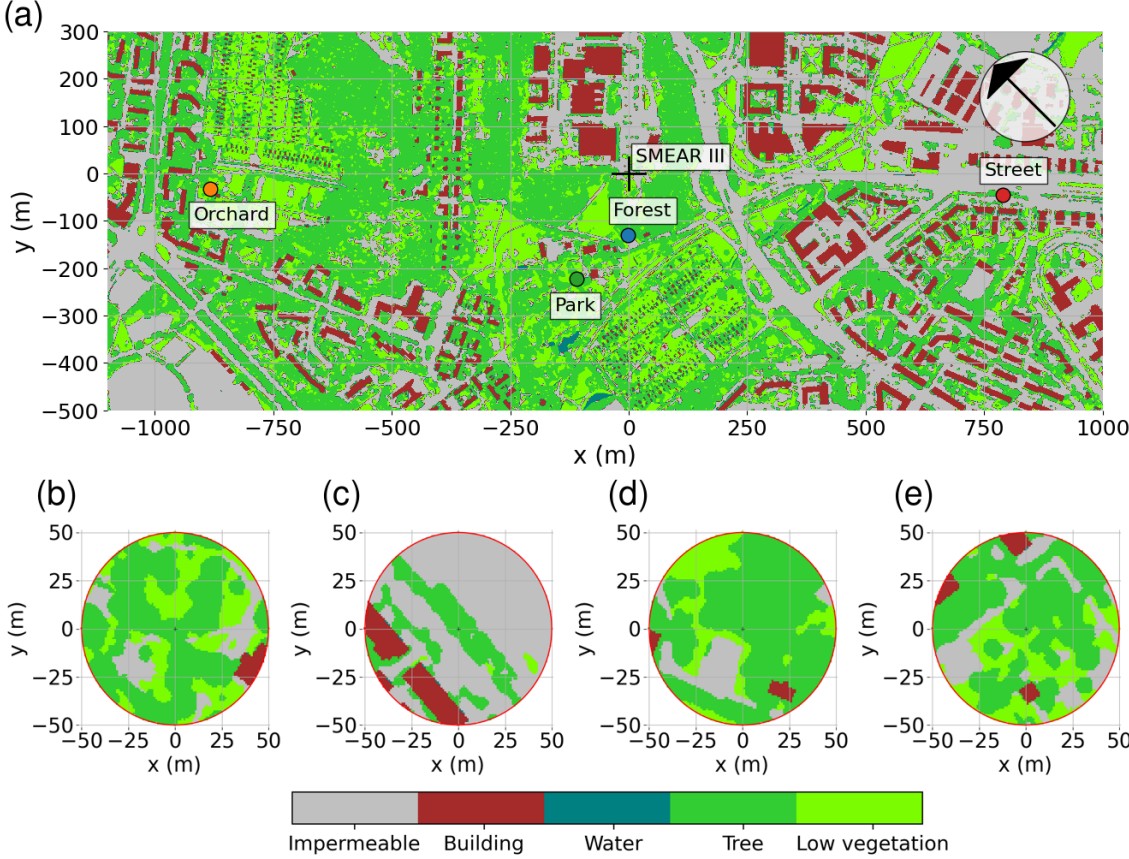

**Figure 1.** (a) Surface cover of the study area in Helsinki (StromJan, 2020) with the location of the monitoring sites, and surface covers at the (b) Park, (c) Street, (d) Forest, and (e) Orchard measurement sites.

**Table 1.** The four study sites, the dominating tree species, their mean diameter at breast height (DBH), sapwood area, height, years since plantation (age) and soil particle type.

|  | Park | Street | Forest | Orchard |
|---|---|---|---|---|
| **Latitude** | 60°12'08.4"N | 60°11'51.6"N | 60°12'07.7"N | 60°12'30.17"N |
| **Longitude** | 24°57'21.4"E | 24°58'13.2"E | 24°57'33.0"E | 24°56'57.77"E |
| **Tree species** | *Tilia cordata* | *Tilia × europaea* | *Betula pendula* | *Malus spp.* |
| **DBH (cm)** | 26.3 | 19.5 | 23.6 | 30 |
| **Sapwood area (cm$^2$)** | 433.8 | 271.9 | 349.7 | 397.0 |
| **Height (m)** | 12.5 | 10 | 22 | 6.5 |
| **Age (years)** | 26 | 34 | 35 | 72 (approx.) |
| **Soil type** | Sand moraine | Fine sand moraine | Sand moraine | Sand clay |





## 2.2 Sap flux measurements

Sap flux measurements were conducted using the Thermal Dissipation Probe (TDP; Granier, 1985). TDP sensors consist of two thermocouple needles (20-30 mm long) where one needle acts as a heating probe and the other as a reference probe. The thermocouples measure the temperature difference between the heated probe and the reference probe, which is used to calculate

the sap flux density and also the whole-tree water use when scaled up. In our study, we selected three sample trees at each site (a total of 12 trees) based on high sun exposure and dominant position at the site. In each tree, a TDP sensor was inserted into the stem xylem at a height of 1.3 m, where a vertical distance of 10 cm was kept between the heated and the reference probe. In the Street and Forest sites, the sensors were installed higher, at 2 m, in order to avoid damage or disturbance. The sensors were installed on the northern side of the stem. To protect and minimize the effect of the thermal gradient, the sensors were

insulated with reflective aluminum foil after installation. The upper probe was heated and the temperature difference ($dT$) was recorded every 1 min using a datalogger (Datataker DT80M). The sap flux density ($J_s$, g cm$^{-2}$h$^{-1}$) was calculated at 1 min interval based on Granier's equation (eqn 1):

$$J_s = 42.84 * ((dT_{max} - dT)/dT)^{1.231},\tag{1}$$

where $dT_{max}$ is the maximum $dT$ where zero $J_s$ was observed. Zero flux condition was based on Lu et al. (2004) where it was

derived as the average of local daily maximum $dT$ of seven consecutive nights. Processing of raw sap flux data was conducted in R (RStudio Team, 2020). Further, daily tree water use was calculated by multiplying the daily sap flux density with the sapwood area. For *Tilia cordata*, *Tilia × europaea* and *Betula pendula*, the sapwood area was derived from the literature using species-specific diameter relationship (Gebauer et al., 2008; Hernandez-Santana et al., 2015) and for Orchard, we calculated the sapwood area by coring the stem of the apple trees. For further analysis, the sap flux density data were selected only for

the growing season (June to September) for both 2020 and 2021. To compare the sap flux density of the four sites, only sunny days data were considered. Sunny days were selected based on the criteria where the daily total $R_g$ was greater than 200 W m$^{-2}$, there was no precipitation, and the mean daytime vapor pressure deficit (VPD) was greater than 0.33 kPa (Riikonen et al., 2016). Sap flux data were averaged to half-hourly, daily and monthly values for each site. In addition, we normalized the sap flux density by dividing $J_s$ with VPD to examine the dependency of $J_s$ on other environmental variables.

## 2.3 Leaf gas exchange

Leaf gas exchange was measured using a portable gas exchange system (Walz GFS-3000, Heinz Walz, GmbH, Germany) with a standard measuring head (8 cm$^2$ cuvette, 2x4 cm). At each site, leaf gas exchange was measured during the summers of 2020 and 2021 with approximately four-week-interval from the same trees as the sap flux was measured. However, at the Park site, one different tree was selected for leaf gas exchange measurements to replace one of the sap flow-equipped tree as it was

difficult to reach the canopy with a manlift. The measurements were made on the southern or southwestern side of each tree and conducted mostly during local morning time (8 AM-12 PM). The measurements were performed on a healthy single leaf. At the Park site, measurements were made at three heights of the canopy (top, middle, and bottom) whereas, at the Street and





Forest sites, only two heights at the top and bottom were monitored. At the Orchard site, only one measurement was made in the middle of the canopy of each tree.

During each measurement, the $CO_2$ level was set to ambient conditions i.e. 415 ppm but the temperature was not set to any value and was rather following the ambient conditions. In case the assimilation at the first 10 min was very low (under 1.5 $\mu$mol m$^{-2}$ s$^{-1}$, a different leaf from the branch was selected and the same measurement steps were repeated. The steps involved setting photosynthetically active radiation (PAR) at 1200 $\mu$mol m$^{-2}$ s$^{-1}$ for 12 minutes and then increasing to a level of 1500 $\mu$mol m$^{-2}$ s$^{-1}$. After reaching the maximum level, PAR gradually decreased down to <1 $\mu$mol m$^{-2}$ s$^{-1}$ over a period of 43 minutes. Altogether, once set, 15 different PAR intensities were included. A simple light response curve was fitted to the net $CO_2$ exchange ($NE$(PAR), $\mu$mol m$^{-2}$ s$^{-1}$), as follows:

$$NE(\mathrm{PAR}) = (A_{max} * \mathrm{PAR})/(\beta + \mathrm{PAR}) - R, \tag{2}$$

where $R$ is the plant respiration i.e. NE measured at PAR = 0 ($\mu$mol m$^{-2}$ s$^{-1}$), $A_{max}$ is the maximum rate of photosynthesis ($\mu$mol m$^{-2}$ s$^{-1}$) and $\beta$ (m$^{-2}$ s$^{-1}$) is the half-saturation constant describing the light intensity where photosynthesis is half of the rate of $A_{max}$.

From the above fitting, only $A_{max}$ is considered in our analysis. Other variables of leaf gas exchange, namely stomatal conductance ($g_s$, mmol m$^{-2}$ s$^{-1}$) and transpiration ($E$, mmol m$^{-2}$ s$^{-1}$) were also recorded during the measurements. Maximum stomatal conductance and transpiration were calculated based on momentary $g_s$ and $E$ at PAR = 1100 $\mu$mol m$^{-2}$ s$^{-1}$.

During the manual measurement campaigns in the summer of 2021, three leaf samples per site were also collected monthly in order to measure their relative water content (RWC). The samples were collected during the late afternoon (4 PM) and the fresh weight (FW) was measured. After that, the leaf samples were soaked overnight and turgid weight (TW) was measured. Later on, the samples were oven dried at 60 °C for 24h and the dry weight (DW) was measured. The RWC was calculated based on the equation below:

$$\mathrm{RWC}(\%) = ((\mathrm{FW} - \mathrm{DW})/(\mathrm{TW} - \mathrm{DW})) * 100, \tag{3}$$

## 2.4 Meteorological and soil data

At all four sites, meteorological variables, including air temperature (Air T, °C) and relative humidity (RH, %) were measured at a height of 1.5-1.8 m with a weather sensor (HMP110, Vaisala, Vantaa, Finland, at Park, Street and Forest sites; HC2A, Rotronic, Bassersdorf, Germany, at Orchard site). Soil sensors (Hydra-probe 2 SDI-12, Stevens, Oregon, USA, except ML3 ThetaProbe, Delta-T, Cambridge, UK, sensors in the Orchard) were installed at 10 and 30 cm depth to measure soil temperature (Soil T, °C) and soil moisture (SM m$^3$ m$^{-3}$). Data were recorded continuously at 1 min intervals and then converted into half-hourly averaged data. Furthermore, the vapor pressure deficit (VPD, kPa) was calculated using Air T and RH based on saturated vapor pressure. Photosynthetically active radiation (PAR, W m$^{-2}$) and precipitation data were collected from the SMEAR III station FI-Kmp measurement tower and roof of a nearby building, respectively (Vesala et al., 2008; Järvi et al., 2009).



## 2.5 Data and statistical analysis

According to Fischer and Schär (2010), the heatwave is defined as a spell of at least six consecutive days with maximum temperatures exceeding the local 90th percentile of the control period. In our study, heatwave (Appendix A1) was defined as the period of consecutive of at least 6 days when the local daily maximum air temperature of the year was exceeding the daily maximum air temperature of the control period (1991-2020). Accordingly, our study period was categorized into heatwave (17 June 2021 to 18 July 2021), pre-heatwave (1 June 2021 to 16 June 2021), post-heatwave (19 July 2021 to 31 August

2021) and no heatwave (1 July 2020 to 31 July 2020) periods. To determine the drought period, Standardised Precipitation-Evapotranspiration Index (SPEI, (Vicente-Serrano et al., 2010) was calculated. Based on the index, June (SPEI = -0.7) and July (SPEI = -0.3) 2021 had mild drought conditions. We separated the measurement period further into dry (22 June 2021 to 27 July 2021) and wet periods (28 July 2021 to 31 August 2021) based on soil moisture (0.17 m$^3$ m$^{-3}$) and daily precipitation (dryperiod has precipitation less than 1 mm).

To test the hypotheses, Kruskal Wallis test followed by Dunn's posthoc test was performed to examine differences in $J_s$ between the sites and different climatic periods (dry/wet, pre-heatwave/heatwave/post-heatwave/no heatwave). First, polynomial regression with 2nd order degree was fitted between daytime mean $J_s$ and daytime daily VPD. Second, multiple linear regression analysis was performed between the $J_s$ and different meteorological variables such as VPD, PAR, soil moisture and soil temperature at 30 cm to determine the effect of the meteorological variables and the relative importance of these mete-

orological variables in controlling the daily $J_s$ was assessed using Student's t-test ($t$) between the individual meteorological and daily $J_s$. Data processing (post-processing of sap flux, meteorological and gas exchange data, statistical analysis) and visualization were conducted in Python (Python version 3).

## 3 Results

### 3.1 Weather conditions

The summer of 2021 (particularly July) was warm and dry, as compared to the summer of 2020, and the climatic reference period (1991 - 2020). The mean air temperature in July 2021 was 21.6°C, being 21% higher than in July 2020 (16.7°C) and 19% higher than the average July during the climatic reference period (18.1°C) (Figure 2a). At the four urban sites, high air temperature (20.2-21.6°C), high soil temperature (15.7-18.5°C), high VPD (0.9 - 1.1 kPa), and low soil moisture (0.1-0.4 m$^3$ m$^{-3}$) were observed during July 2021 (Figure 3 for Park site, Appendices A2, A3 and A4 for other sites). The total precipitation

for the months of June and July in 2021 (86 mm) was 51% lower than during June and July 2020 (177 mm) and 27% lower than on average for the climatic reference period (117 mm) (Figure 2b).

The microclimatic conditions varied across the four sites during summer 2021. Higher (8%) mean air temperature was observed at both Orchard (19.2 °C) and Street sites (19.3 °C) as compared to Forest (17.9 °C) and Park (17.8 °C) sites (Table 2). Similar VPD was found at the Orchard (0.78 kPa), Park (0.75 kPa) and Forest sites (0.75 kPa), but 11% higher VPD at the

Street site (0.84 kPa). Also, the mean soil temperature at the Street site (19.9°C) was 24-34% higher than at the Park (16.1°C),


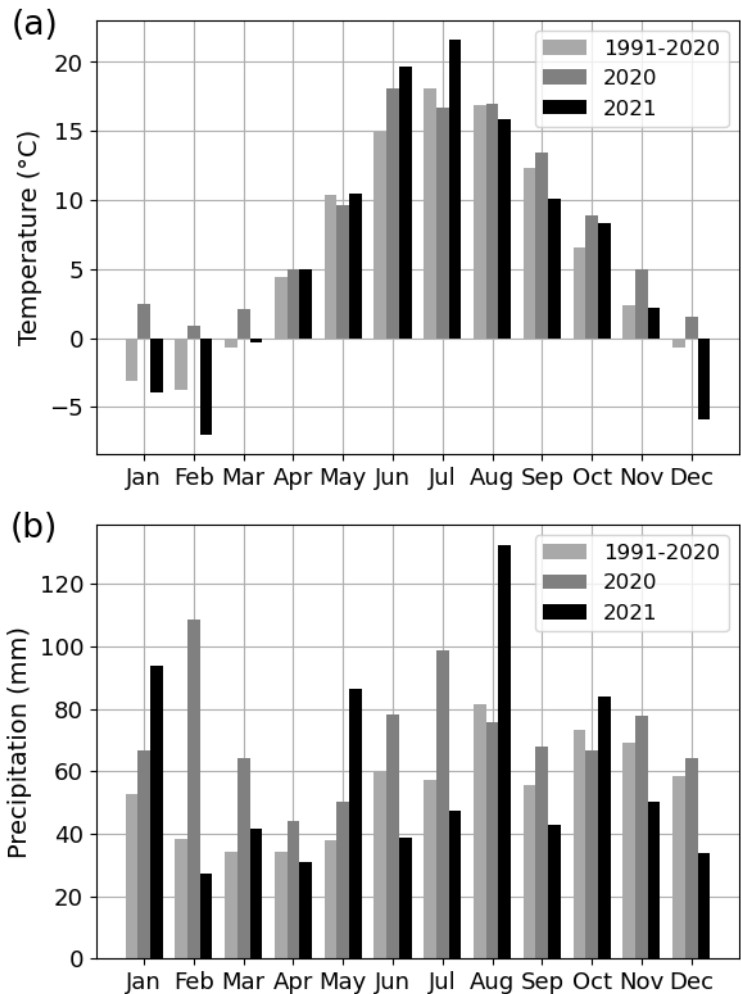

**Figure 2.** (a) Monthly mean air temperature and (b) monthly total precipitation for the year 2020 and 2021, and the climatic reference period of 30 years (1991-2020).

Orchard (15.2°C) and Forest sites (14.8°C). Soil moisture varied largely as the Orchard site had the highest (0.37 m³ m⁻³) and the Forest site the lowest (0.09 m³ m⁻³) wheareas Park and Street sites had 0.13 and 0.22 m³ m⁻³ respectively. Soil moisture conditions were recovered after the rainfall at all the sites except the Forest site, where soil moisture remained low throughout the late summer after the hot and dry July 2021 (Appendix A3).

## 3.2 Variability in tree water use in 2021

Based on sunny days, the mean daily water use of the trees in the Park, Street, Forest and Orchard were $0.32 \pm 0.01$ kg cm⁻² day⁻¹, $0.42 \pm 0.01$ kg cm⁻² day⁻¹, $0.20 \pm 0.01$ kg cm⁻² day⁻¹ and $0.46 \pm 0.01$ kg cm⁻² day⁻¹, respectively and differed



**Figure 3.** Meteorological and soil data from 2021 showing hourly a) air temperature (Air T), b) water vapor deficit (VPD), c) soil temperature (Soil T) and d) soil moisture measured at the Park site, and e) daily mean Photosynthetically active radiation (PAR) and daily sum precipitation data measured at the SMEARIII station. The orange shading indicates the heatwave period during the summer of 2021 and the black vertical line indicates the onset of the wet period. The yellow markers in panel (c) denote the dates of manual leaf gas measurements.

significantly between the sites (Figure 4a) (P <0.05). The mean daytime sap flux density ($J_s$) for the summer period (June-August) was highest at the Orchard site with $20.6 \pm 0.3$ g cm$^{-2}$ h$^{-1}$, lowest with $8.1 \pm 0.1$ g cm$^{-2}$ h$^{-1}$ at the Forest site, and

$14.4 \pm 0.2$ g cm$^{-2}$ h$^{-1}$ and $17.7 \pm 0.3$ g cm$^{-2}$ h$^{-1}$ at the Park and Street sites, respectively. The monthly mean $J_s$ differed significantly between the sites (P<0.05) (Figure 4b).





**Table 2.** Monthly mean air temperature (Air T, °C), water vapor pressure deficit (VPD, kPa), soil temperature (Soil T, °C) and soil moisture ($m^3$ $m^{-3}$) for the four study sites in 2021.

|         |           | Air T | VPD  | Soil T | Soil moisture |
|---------|-----------|-------|------|--------|---------------|
| **Park**    | **June**      | 18.4  | 0.84 | 14.4   | 0.16          |
|         | **July**      | 20.2  | 0.96 | 17.5   | 0.09          |
|         | **August**    | 14.8  | 0.45 | 14.7   | 0.15          |
|         | **September** | 8.7   | 0.40 | 11.0   | 0.19          |
| **Street**  | **June**      | 19.8  | 0.93 | 18.5   | 0.23          |
|         | **July**      | 21.6  | 1.07 | 22.8   | 0.25          |
|         | **August**    | 16.1  | 0.51 | 18.5   | 0.25          |
|         | **September** | 10.0  | 0.40 | 12.7   | 0.29          |
| **Forest**  | **June**      | 18.0  | 0.77 | 13.1   | 0.16          |
|         | **July**      | 20.7  | 1.01 | 16.9   | 0.07          |
|         | **August**    | 15.2  | 0.48 | 14.5   | 0.06          |
|         | **September** | 8.7   | 0.34 | 11.2   | 0.06          |
| **Orchard** | **June**      | 21.1  | 0.81 | 15.5   | 0.44          |
|         | **July**      | 21.1  | 1.05 | 15.7   | 0.28          |
|         | **August**    | 15.7  | 0.49 | 14.6   | 0.40          |
|         | **September** | 9.4   | 0.35 | 12.9   | 0.49          |

### 3.3 The effect of heatwave on sap flux density

During heatwave period, the mean daily water use of the trees at the Park, Street, Forest and Orchard sites were $0.38 \pm 0.02$ kg $cm^{-2}$ $day^{-1}$, $0.42 \pm 0.02$ kg $cm^{-2}$ $day^{-1}$, $0.24 \pm 0.01$ kg $cm^{-2}$ $day^{-1}$ and $0.52 \pm 0.02$ kg $cm^{-2}$ $day^{-1}$, respectively. At the

Park site, the mean $J_s$ during heatwave period was 59% higher than during no heatwave period and 39% higher than during post-heatwave period but there was no significant difference with the pre-heatwave period. At the Street site, there was no significant difference in the mean $J_s$ between heatwave, no heatwave, pre-heatwave and post-heatwave periods. At the Forest site, the mean $J_s$ during heatwave period was 13% higher than during pre-heatwave and 67% higher than during post-heatwave periods. At the Orchard site, the mean $J_s$ during heatwave period was 35% higher than during post-heatwave period but there

was no significant difference with pre-heatwave period (Figure 5a, P<0.05). At the Forest and Orchard sites, data from no heatwave period were not available.

When normalized with VPD, there were no significant differences in the $J_s$ between heatwave, no heatwave and pre-heatwave and post-heatwave periods in the Park, Forest or Orchard (Figure 5b). At the Street site, the normalized $J_s$ during heatwave period was 33% lower than during no heatwave and 7% lower than during pre-heatwave but it did not differ from

that of post-heatwave period.



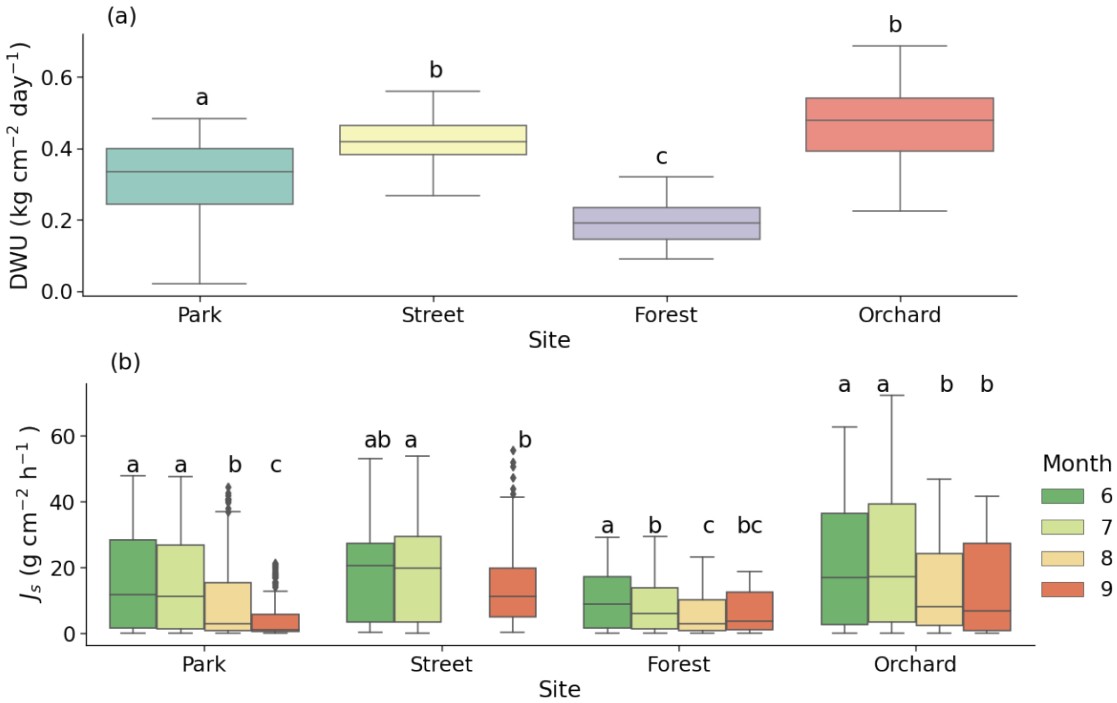

**Figure 4.** (a) Daily water use (DWU) of the trees and (b) monthly mean sap flux density ($J_s$) at the four urban vegetation sites: Park (*Tilia cordata*) , Street (*Tilia × europaea*), Forest (*Betula pendula*) and Orchard (*Malus spp.*). The letters indicate the significant differences between the (a) sites and (b) monthly mean $J_s$ at each site (P<0.05).

### 3.4 The effect of drought on sap flux density

During dry period, the mean daily water use at the Park, Street, Forest and Orchard site was $0.36 \pm 0.02$ kg cm$^{-2}$ day$^{-1}$, $0.42 \pm 0.01$ kg cm$^{-2}$ day$^{-1}$, $0.20 \pm 0.01$ kg cm$^{-2}$ day$^{-1}$ and $0.50 \pm 0.02$ kg cm$^{-2}$ day$^{-1}$, respectively. The mean $J_s$ was significantly higher during dry period than during wet period at all sites (Figure 6a, P<0.05). At the Park, Street, Forest and
Orchard sites, the mean $J_s$ was 66%, 31%, 43% and 53%, respectively, higher during dry period than wet period.

The normalized $J_s$ was significantly lower during dry than wet period at all sites, with 16% lower at the Park site, 48% lower at the Street site, 28% lower at the Forest site and 26% lower at the Orchard site (Figure 6b).

### 3.5 Leaf gas exchange during the heatwave and drought periods

We compared leaf gas exchange variables, namely $A_{max}$, $g_s$ and $E$, between the different heatwave periods and between dry
and wet periods (Table 3). No significant differences in these three variables were found at the Park, Forest and Orchard sites between the different heatwave periods; however, at the Street site, $A_{max}$ and $E$ were significantly (P<0.05) higher during





**Table 3.** Average of leaf gas exchange variables: Maximum assimilation ($A_{max}$, $\mu$mol m$^{-2}$ s$^{-1}$), Stomatal Conductance ($g_s$, mmol m$^{-2}$ s$^{-1}$), Transpiration (E, mmol m$^{-2}$ s$^{-1}$) during heatwave, no heatwave, pre-heatwave and post-heatwave periods and during dryand wet periods at the four sites. The letters indicate the significant difference between the various heatwave periods or drought periods.

| Site | Type / Period | $A_{max}$ | $g_s$ | E |
|------|---------------|-----------|-------|---|
| **Park** | heatwave | $13.6 \pm 1.3$ | $112.1 \pm 14.1$ | $1.5 \pm 0.1$ |
| | pre-heatwave | $13.6 \pm 1.3$ | $123.5 \pm 0$ | $1.9 \pm 0$ |
| | post-heatwave | $15.3 \pm 1.1$ | $141.3 \pm 11$ | $1.1 \pm 0.1$ |
| | no heatwave | $17.2 \pm 1.4$ | $147.8 \pm 15.5$ | $1.5 \pm 0.1$ |
| | dry | $15.0 \pm 0.8$ | $114.5 \pm 7.7^a$ | $1.5 \pm 0.1$ |
| | wet | $14.3 \pm 2.0$ | $163.3 \pm 17.5^b$ | $1.7 \pm 0.2$ |
| **Street** | heatwave | $10.9 \pm 1.0^a$ | $91.4 \pm 7.3$ | $1.2 \pm 0.1^a$ |
| | pre-heatwave | - | - | - |
| | post-heatwave | $6.6 \pm 0.8^b$ | $64.2 \pm 12.0$ | $0.8 \pm 0.1^b$ |
| | no heatwave | $8.7 \pm 1.2^a$ | $73.0 \pm 11.1$ | $0.8 \pm 0.1^a$ |
| | dry | $10.9 \pm 1.0^a$ | $91.4 \pm 7.3$ | $1.2 \pm 0.1$ |
| | wet | $6.6 \pm 0.8^b$ | $64.2 \pm 12.0$ | $0.8 \pm 0.1$ |
| **Forest** | heatwave | $16.4 \pm 1.0$ | $105.7 \pm 17.8$ | $1.3 \pm 0.2$ |
| | pre-heatwave | $16.3 \pm 0$ | $128.4 \pm 0$ | $1.5 \pm 0$ |
| | post-heatwave | $10.7 \pm 2.9$ | $104.2 \pm 31.2$ | $1.1 \pm 0.3$ |
| | no heatwave | $17.4 \pm 2.3$ | $133.5 \pm 12.1$ | $1.4 \pm 0.1$ |
| | dry | $16.4 \pm 1.0$ | $105.7 \pm 17.8$ | $1.3 \pm 0.2$ |
| | wet | $10.7 \pm 2.9$ | $104.2 \pm 31.2$ | $1.1 \pm 0.3$ |
| **Orchard** | heatwave | $14.1 \pm 0.7$ | $126.8 \pm 5.4$ | $0.7 \pm 0.2$ |
| | pre-heatwave | $13.3 \pm 0$ | $125.3 \pm 0$ | $2.0 \pm 0$ |
| | post-heatwave | $13.6 \pm 1.0$ | $157.3 \pm 12.9$ | $1.8 \pm 0.1$ |
| | no heatwave | - | - | - |
| | dry | $13.1 \pm 1.4$ | $135.4 \pm 13.6$ | $1.6 \pm 0.1$ |
| | wet | $14.0 \pm 0.9$ | $170.9 \pm 20.4$ | $1.9 \pm 0.2$ |





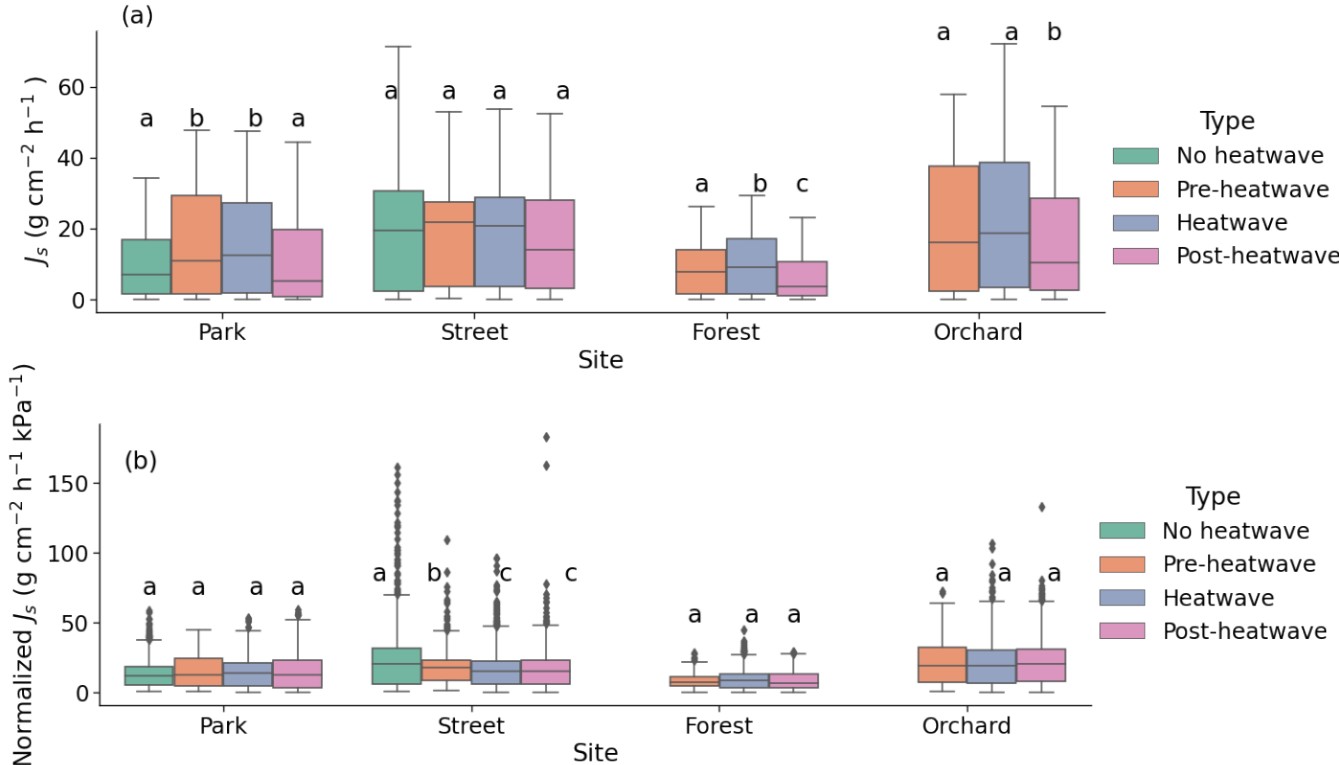

**Figure 5.** The sap flux densities at the four urban vegetation sites during different periods regarding a heatwave. a) mean $J_s$ of the whole day (b) mean normalized $J_s$ by VPD on sunny days. The letters indicate the significant differences between the different periods within each site (P<0.05).

heatwave than post-heatwave periods but there was no significant difference as compared to no heatwave period. Also, $g_s$ showed no difference between the different heatwave periods at the Street site.

Comparing leaf gas exchange variables between dry and wet periods, we found that at the Street site, $A_{max}$ was significantly
higher (P<0.05) during dry period than wet period but no significant differences in $g_s$ and $E$ between dry and wet periods were found. Also, at the Park site, $g_s$ was found to be significantly lower (P<0.05) during dry period than wet period. At the Forest and Orchard sites, no significant differences were found in $A_{max}$, $g_s$ and E between dry and wet periods.

The monthly relative leaf water content (RWC) as a proxy of leaf water potential showed that RWC was found to be lower (4-35 %) during July as compared to June and August at the Forest and Orchard sites; however, RWC was found to be higher
(5-8 %) during July than the other summer months at Park and Street sites (Appendix A5).



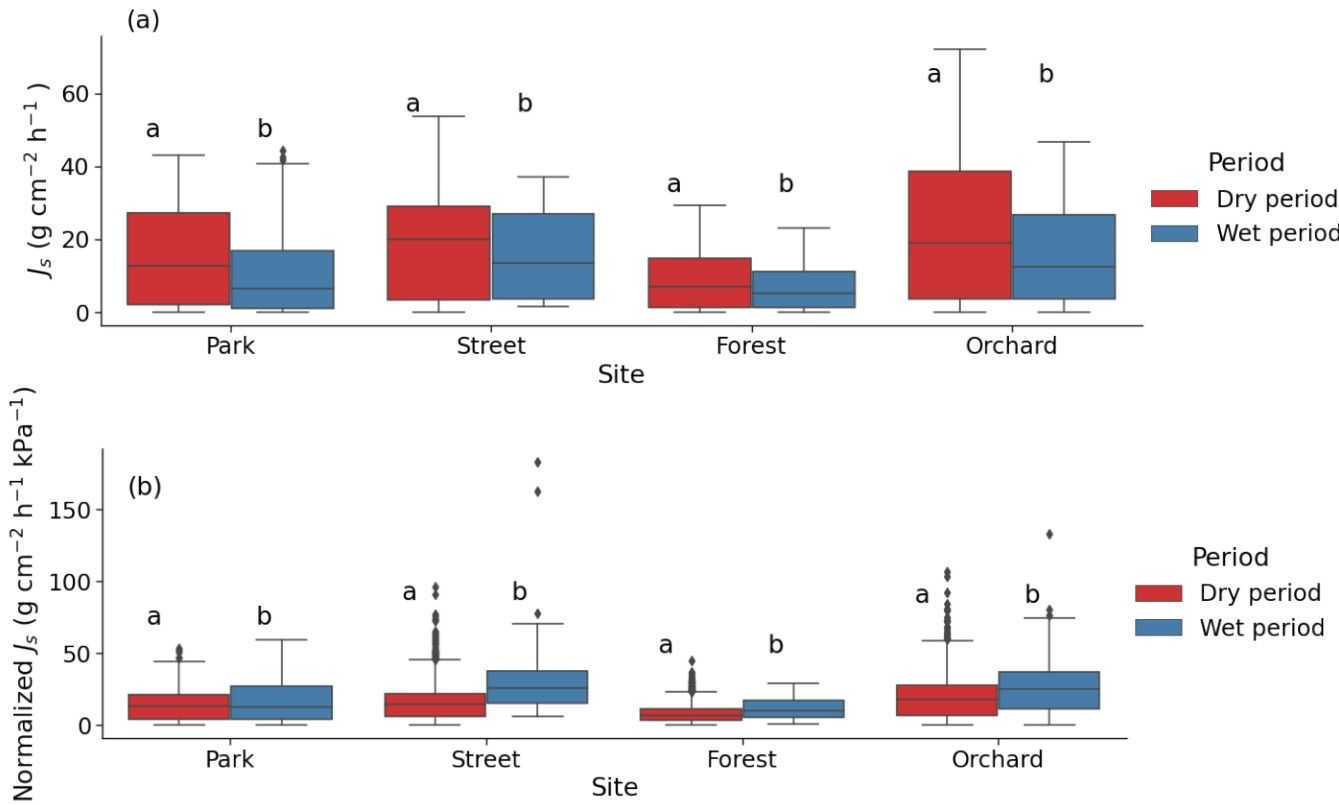

**Figure 6.** Sap flux density ($J_s$) during dry and wet periods at the four study sites. Based on sunny days, (a) data represented mean $J_s$ of the whole day and (b) normalized $J_s$ by VPD. The letters indicate the significant differences $J_s$ between the dry and wet period at each site (P<0.05)

### 3.6 Environmental control on sap flux density

We tested the relationship between the daily daytime mean VPD and $J_s$ using 2nd order polynomial regression (Figure 7, Table 4). VPD explained the variation in $J_s$ less during the heatwave than during post-heatwave and pre-heatwave except for Forest where the VPD was not a significant driver at all during pre-heatwave. During wet period, VPD explained a higher share of the variation of $J_s$ except for Street where the wet data coverage is very low (Fig. 7).

Multiple linear regression between $J_s$ and a higher number of environmental variables (VPD, PAR, Soil T and SM) showed that PAR and VPD were the significant drivers for $J_s$ at the Park, Street and Orchard site but PAR was the only significant driver at the Forest site (Table 5). In addition, soil moisture was a significant variable in certain circumstances at certain sites such as in Park during heatwave and dry periods and in Forest, during post-heatwave, dry and wet periods i.e. in the latter part of the growing season in general.



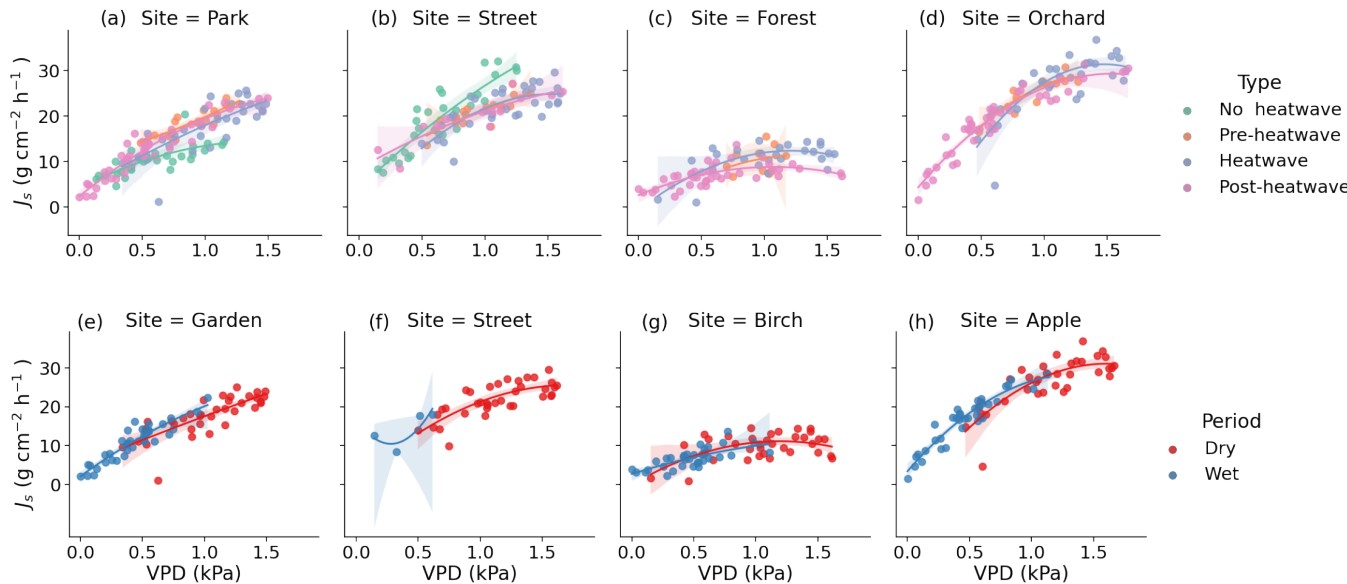

**Figure 7.** Observed (dots) and modeled (lines) relationship between the daytime mean daily VPD and $J_s$ during the heatwave periods and both dry and wet periods. Panel (a),(b),(c),(d) show the relationship between VPD and $J_s$ at the Park, Street, Forest and Orchard site during the various heatwave periods. Panel (e),(f),(g),(h) show the relationship between VPD and $J_s$ at the Park, Street, Forest and Orchard site during the dry and wet periods. The model is 2nd order polynomial fit (see Table 4).

**Table 4.** Relationship between vapor pressure deficit and daily mean sap flux density using 2nd order polynomial fit (see Fig. 7). The values are showing the squared coefficient of correlation (adj $R^2$) between the two variables. Relationships statistically significant at 0.05, 0.01 and 0.001 levels are marked with *, ** and *** respectively and non-significant as ns.

|  | **no heatwave** | **pre-heatwave** | **heatwave** | **post-heatwave** | **dry** | **wet** |
|---|---|---|---|---|---|---|
| **Park** | 0.53*** | 0.73*** | 0.57*** | 0.90*** | 0.60*** | 0.85*** |
| **Street** | 0.80*** | 0.62*** | 0.55*** | 0.75*** | 0.58*** | $0.22^{ns}$ |
| **Forest** | - | $0.16^{ns}$ | 0.43*** | 0.44*** | 0.25** | 0.48*** |
| **Orchard** | - | 0.77*** | 0.69*** | 0.91*** | 0.67*** | 0.90*** |

# 4   Discussion

In this study, we assessed the response of urban tree water use and leaf gas exchange to heat and drought during two contrasting summers 2020 and 2021, the latter being hot and dry, across four urban green areas in Helsinki. The results showed that tree water use, measured with sap flux density increased during the heatwave and the dry period but carbon assimilation, stomatal conductance and transpiration at leaf level did not generally change during these various periods. The increase in sap flux density during hot and dry periods varied across the urban sites that had different tree species and growing conditions. VPD




**Table 5.** Multiple linear regression between daily mean sap flux density and various environmental drivers (VPD, PAR, Soil temperature and Soil moisture) as independent variables. t stat value indicates the relative importance of the variables in controlling the daily sap flux variations. Relationships statistically significant at 0.05, 0.01 and 0.001 levels are marked with *, ** and *** respectively and non-significant as ns.

| Site | | All data | no heatwave | pre-heatwave | heatwave | post-heatwave | dry period | wet period |
|---|---|---|---|---|---|---|---|---|
| **Park** | $R^2$ | 0.63 | 0.59 | 0.75 | 0.74 | 0.93 | 0.74 | 0.92 |
| | Intercept | $5 (1.4)^{ns}$ | $-1.6 (-0.2)^{ns}$ | $-16.6 (-0.9)^{ns}$ | $-25.15 (-1.6)^{ns}$ | $4.1 (1.1)^{ns}$ | $-13.9(-1.1)^{ns}$ | $3.0 (0.7)^{ns}$ |
| | VPD | $9 (7.6)^{***}$ | $6.2 (3.1)^{***}$ | $9.1 (3.5)^{***}$ | $10.4 (5.1)^{***}$ | $9.3 (8.0)^{***}$ | $9.5 (5.2)^{***}$ | $11.0 (5.7)^{ns}$ |
| | PAR | $0.01 (3.5)^{***}$ | $0.01 (1.14)^{ns}$ | $0.01 (1.4)^{ns}$ | $0.03 (3.1)^{***}$ | $0.02 (5.4)^{***}$ | $0.02 (3.6)^{***}$ | $0.01 (4.1)^{**}$ |
| | Soil T | $-0.1 (-0.3)^{ns}$ | $0.5 (1.1)^{ns}$ | $1 (1.4)^{ns}$ | $0.8 (1.32)^{ns}$ | $-0.1 (-0.6)^{ns}$ | $0.4 (0.7)^{ns}$ | $0.03 (0.1)^{ns}$ |
| | SM | $-9.4 (-1.0)^{ns}$ | $-10.5 (-0.4)^{ns}$ | $45.6 (1.0)^{ns}$ | $79.6 (2.3)^{***}$ | $-8.0 (-1.0)^{ns}$ | $48.1 (2.1)^{***}$ | $-15.9 (-2.2)^{ns}$ |
| **Street** | $R^2$ | 0.73 | 0.84 | 0.65 | 0.71 | 0.75 | 0.71 | - |
| | Intercept | $5.9 (2.27)^{*}$ | $6.2 (0.3)^{ns}$ | $7.0 (1.0)^{ns}$ | $10.6 (1.1)^{ns}$ | $40.7 (1.3)^{ns}$ | $22.5 (2.0)^{ns}$ | - |
| | VPD | $9.3 (7.8)^{**}$ | $17.5 (6.4)^{***}$ | $7.6 (2.5)^{*}$ | $4.6 (2.6)^{*}$ | $6.1 (1.4)^{ns}$ | $5.9 (3.4)^{***}$ | - |
| | PAR | $0.03 (7.1)^{**}$ | $0.02 (2.5)^{*}$ | $0.02 (1.7)^{ns}$ | $0.03 (4.4)^{***}$ | $0.01 (1.0)^{ns}$ | $0.03 (4.0)^{***}$ | - |
| | Soil T | $-0.3 (-3.3)^{**}$ | $-0.5 (-0.9)^{ns}$ | $0.07 (0.2)^{ns}$ | $-0.1 (-0.5)^{ns}$ | $-1.1 (-1.2)^{ns}$ | $-0.7 (-1.6)^{ns}$ | - |
| | SM | $12.6 (2.4)^{*}$ | $12.0 (0.2)^{ns}$ | $-9.7 (-0.7)^{ns}$ | $-13.9 (-1.0)^{ns}$ | $-34.5(-0.5)^{ns}$ | $-8.3 (-0.8)^{ns}$ | - |
| **Forest** | $R^2$ | 0.56 | - | 0.93 | 0.67 | 0.68 | 0.63 | 0.72 |
| | Intercept | $2.4 (1.4)^{ns}$ | - | $91.0 (2.4)^{ns}$ | $-2.3 (-0.1)^{ns}$ | $-16.2(-1.8)^{ns}$ | $-17.7(-2.1)^{*}$ | $-17.3(-1.5)^{ns}$ |
| | VPD | $2.1 (2.0)^{ns}$ | - | $2.9 (1.5)^{ns}$ | $3.5 (1.7)^{ns}$ | $-0.3 (-0.3)^{ns}$ | $2.9(1.9)^{ns}$ | $0.1(0.03)^{ns}$ |
| | PAR | $0.02 (5.1)^{***}$ | - | $0.01 (1.3)^{ns}$ | $0.02 (3.4)^{***}$ | $0.02(4.99)^{***}$ | $0.02(3.6)^{***}$ | $0.02(3.5)^{***}$ |
| | Soil T | $-0.1 (-1.1)^{ns}$ | - | $-1.0 (-0.9)^{ns}$ | $-0.6 (-0.8)^{ns}$ | $-0.8 (-4.4)^{***}$ | $0.01(0.03)^{ns}$ | $-0.8(-3.4)^{**}$ |
| | SM | $-1.4 (-0.2)^{ns}$ | - | $-345.0 (-3.1)^{*}$ | $137.7 (1.1)^{ns}$ | $515.0(3.6)^{***}$ | $222.1(5.0)^{***}$ | $515.2(2.8)^{**}$ |
| **Orchard** | $R^2$ | 0.89 | - | 0.86 | 0.77 | 0.93 | 0.74 | 0.92 |
| | Intercept | $9.8 (1.7)^{ns}$ | - | $-88.2 (-2.3)^{ns}$ | $57.9 (0.8)^{ns}$ | $27.0 (2.2)^{*}$ | $-31.2(-1.4)^{ns}$ | $3.1 (0.8)^{ns}$ |
| | VPD | $9.2 (8.0)^{***}$ | - | $5.9 (1.9)^{ns}$ | $5.4 (2.1)^{ns}$ | $7.8 (5.4)^{***}$ | $7.1 (3.4)^{**}$ | $11 (5.8)^{***}$ |
| | PAR | $0.04 (8.2)^{***}$ | - | $0.03 (2.9)^{*}$ | $0.05 (4.5)^{***}$ | $0.04 (7.6)^{***}$ | $0.05 (4.9)^{***}$ | $0.02 (4.07)^{***}$ |
| | Soil T | $-0.5 (-1.5)^{ns}$ | - | $0.6 (0.5)^{ns}$ | $-2.8 (-0.7)^{ns}$ | $-1.6 (-2.1)^{*}$ | $2.1 (1.6)^{ns}$ | $0.03 (0.1)^{ns}$ |
| | SM | $1.5 (0.4)^{ns}$ | - | $160.3 (3.2)^{*}$ | $-32.3 (-1.5)^{ns}$ | $-2.9 (-0.8)^{ns}$ | $-1.6 (-0.2)^{ns}$ | $-15.9(-2.2)^{***}$ |

Values indicate coefficient $(t)^{sig}$





was the main driver of tree water use during the heatwave and dry period. However, the VPD explained less of the variation in the transpiration during the heatwave and dry period as compared to other periods; giving support to our hypothesis H3. Overall, it seems that the hot and dry conditions were not severe enough to trigger notable physiological adaptation as the urban trees

in our study continued to function typically during the summer of 2021. Even though the air temperature was notably higher and precipitation notably lower than during previous summers, we did not get full support for the hypotheses H1 and H2 and conclude that the severe weather events did not alter the stomatal action and therefore the observed photosynthetic potential. However, we found some interesting insights that we will discuss further in this chapter.

## 4.1 Site variability

In our study, we observed that the four urban vegetation sites in Helsinki exhibit variable microclimatic conditions where air temperature under the canopy, soil temperature and soil moisture were different. The highest air and soil temperatures were measured at the Street site, where impervious surfaces increase the temperatures due to heat storage. The high air temperature at the Orchard site is due to exposure of the site to direct sunlight throughout the day heating the garden. Also, the high soil moisture at the Orchard site is mainly due to the difference in soil type where the sand clay in Orchard has higher water holding

capacity than sand moraine soil type at the other three sites. VPD was similar at all other sites except the Street site, where it was clearly higher. We speculate that the highest deficit at the Street site is again due to the large fraction of impervious surfaces (Whitlow et al., 1992; Kjelgren and Montague, 1998). Similar variability of meteorological conditions between different urban forests were found in Los Angeles metropolitan city; where the urban forest located near the city were warmer with high VPD and lower photosynthetically active radiation as compared to the urban forests located closer to the coast (Pataki et al., 2011).

Among 10 different tree species in the city of Basel (Switzerland), the tree crown temperature was lower in the park than the street; reporting that it is species-specific for the cooling effect of urban trees (Leuzinger et al., 2010). The difference in microclimatic conditions has been observed to vary depending on the type of vegetation, the composition of the species, the amount of green cover and the impervious surface in urban vegetation (Perini and Magliocco, 2014; Kjelgren and Clark, 1992).

We observed that tree water use varied significantly across the four urban sites; Orchard and Street sites had the highest tree

water use during the summer of 2021 and it was the lowest at the Forest site. The higher water use at the Orchard and Street sites was mainly due to site conditions such as high soil moisture and tree characteristics such as bigger stem size. Water availability at the Street and Orchard (0.23-0.49 $m^3$ $m^{-3}$) was higher as compared to the Park and Forest (0.06-0.19 $m^3$ $m^{-3}$). Other studies have also shown high variability in tree water use across different urban green areas (Pataki et al., 2011; McCarthy and Pataki, 2010; Sushko et al., 2021). For example in the high-latitude city of Gothenburg (Sweden), *T. europaea* had two times higher

daytime transpiration rates in a park compared to a street site (Konarska et al., 2016). The found differences in tree water use between the sites in our study can also be related to different tree species growing at the sites. The water use of *Tilia Cordata* at the Park was lower than that of *Malus sp* at the Orchard and *Tilia × europaea* at the Street but higher than that of *Betula Pendula* at the Forest site. The low tree water use of *Betula Pendula* at the Forest site could be reduced due to the rather strong stomatal control typical for *Betula Pendula*, i.e. it closes the stomata easily during dry conditions, whereas *Tilia Cordata* found

at the Park site has less sensitive stomatal control, i.e. it keeps stomata open even in mild drought as it can tolerate drought





better than *Betula Pendula*. Differences in water use between different urban tree species have also been reported before. Tree water use of *Tilia Cordata* Mill. were reported to be three times higher than water use of *Robinia pseudoacacia* L. tree in the streets of Munich, Germany (Rahman et al., 2019) and water use of *Tilia × vulgaris* were found to be 1/4 lower than water use of *Alnus glutinosa* in the streets of Helsinki, Finland (Riikonen et al., 2016). In general, the daily tree water use found in this
latter study was lower (in the years 2008-2011) compared to the water use of urban trees we found at the four studied sites in Helsinki. Overall, the tree water use across different urban sites varies not only due to site and climatic conditions but also due to the planted tree species.

## 4.2 Transpiration rate and leaf gas exchange during drought and heatwave

We observed varying responses of tree water use during heatwave and drought periods at the studied sites. At the Park, Forest
and Orchard sites, $J_s$ increased by 35-67% during the heatwave compared to periods of no heatwave and/or post-heatwave, whereas the heatwave did not affect $J_s$ in the Street site. Pre-heatwave period did not differ from the heatwave period in terms of $J_s$ in the Park and Orchard sites, but there was a small increase (13%) at the Forest site during the heatwave period. During the dry period, $J_s$ was significantly higher than during wet period at all sites. The leaf gas exchanges such as $A_{max}$, $g_s$ and $E$ did not change or reduce during the heatwave and dry period; thus, indicating no changes in the photosynthetic potential
during these periods. Hence, we conclude that the weather was yet not severe enough to support the hypotheses H1 and H2 in our study

VPD is the driving force for transpiration, so an increase in VPD leads to an increase in transpiration and tree water use unless stomata in the leaves close to limit transpiration. The ratio between $J_s$ and VPD significantly decreased due to drought and the relative importance of VPD explaining the observed variation in sap flow was lower in dry and heatwave conditions compared
to other periods; giving support to our hypothesis H3. In theory, these results could indicate, that the trees limited their water transport via stomatal control in harsh conditions. However, that was not clearly captured by the leaf-level measurements which on the other hand, may not represent the conditions over the different periods as well as these automatic measurements. Moreover, the leaf-level measurements were based on single-day measurements, which might not fully cover the heatwave or drought periods in our study. Together these results indicate that the observed increase in transpiration at the studied sites was
caused by an increase in the driving force for transpiration, VPD. Similar responses were also observed in the city of Dresden, Germany where the transpiration rate and maximum stomatal conductance of *Tilia cordata* and *Corylus colurna* were largest during dry summer months in 2013 (Gillner et al., 2015a).

Stomatal control limits plant transpiration during drought. For example, Rötzer et al. (2021) found a substantial reduction of 63 % in the transpiration rate in urban *Tilia cordata* and *Robinia pseudoacacia* trees in the city of Würzburg, Germany
during the European drought in 2018. During the dry period of this study, soil moisture decreased by 18-62% at the four sites as compared to wet period. During heatwave period, soil moisture decreased by 30-58% as compared with the pre-heatwave period at all sites except for the Street site. The availability of soil moisture at the Street site allowed an increase in $A_{max}$ and $E$ during the heatwave period. However, the observed reductions in soil moisture did not seem to be enough to cause strong stomatal regulation of transpiration (i.e. no change in $g_s$ except for the Park site and no reduction in $J_s$). Previous studies





have also reported that transpiration during the local extreme high temperature was maintained when there was sufficient water availability in the soil in different urban green sites in Los Angeles Metropolitan, US (Pataki et al., 2011). Similarly, previous studies have shown that VPD is a significant driver for sap flow in *Tilia × vulgaris* street trees in Helsinki (Riikonen et al., 2016), and VPD and solar radiation for daytime transpiration rates in seven different tree species, including *Betula pendula*, in Gothenburg, Sweden (Konarska et al., 2016). Similarly to our results regarding the Park site, Konarska et al. (2016) also found that the maximum stomatal conductance was reduced by 50% in the studied species in Gothenburg even though transpiration rate remained high during dry conditions compared to wet conditions.

During the heatwave and the dry period, VPD explained most of the variation in the daily mean $J_s$ at the Park (57-60%), Street (55-58%) and Orchard sites (62-69%). Similarly, previous studies in high latitude cities have reported that VPD correlates well with $J_s$ in street trees (adj. $R^2 = 0.74$) in Helsinki, Finland (Riikonen et al., 2016), and in urban trees ($R^2$=0.44-0.75) in Gothenburg, Sweden (Konarska et al., 2016). Also in Boston, Massachusetts, VPD has been shown to correlate with $J_s$ ($R^2$ = 0.63, (Winbourne et al., 2020)). However, VPD did not explain daily variation in $J_s$ at the forest site of our study. Also, $J_s$ saturated after reaching certain VPD levels, especially at Forest and Orchard sites, and the saturation took place already in rather low VPD levels in the case of the Forest site compared to the Orchard site. The difference in saturation levels may be species-specific or caused by differences in the availability of soil moisture as the Orchard site had higher soil moisture than the Forest site during the heatwave and dry period. *Tilia* at the Park and Street sites seemed to be more linearly correlated with VPD, suggesting that transpiration continues to increase with increasing VPD during the stressful periods at these sites. With sufficient water supply by irrigation, less saturation with VPD was observed in urban trees previously (Winbourne et al., 2020; Marchin et al., 2022). Similarly, the non-saturation of $J_s$ with high VPD observed at the Park site may be due to irrigation during the dry period. Several previous studies in urban trees have shown that the relationship between VPD and transpiration are species-specific and it is more typical that VPD increases linearly in diffuse-porous trees but saturates in ring-porous trees (Bush et al., 2008; Rahman et al., 2019). All the species studied here are diffuse-porous.

In addition to VPD, $J_s$ was also explained by other environmental variables. The relative importance of soil moisture, soil temperature and solar radiation in explaining $J_s$ differed significantly between the four studied urban sites. In addition to VPD, PAR was among the main environmental drivers of $J_s$ at Park, Orchard and Street sites, whereas PAR alone explained $J_s$ at the Forest site (Table 5). Soil temperature was significantly related with $J_s$ only at the Street site. When the different climatic periods were analyzed separately, soil conditions (moisture and/or temperature) affected $J_s$ during heatwave ($t = 2.3$, P <0.05) and dry periods (($t = 2.1$, P <0.05) at the Park site, but not at the Street, Forest and Orchard sites. This can be explained by the irrigation provided at the Park site during the dry period. Overall, the environmental variables that best explained urban tree transpiration during hot and dry conditions were VPD and PAR, and the relative importance of these two varied depending on the tree species, growing conditions and irrigation practices.

The high $J_s$ during the heatwave in the studied green areas suggests transpirational cooling. Trees at the Orchard and Street sites had the highest transpiration rates and trees in the Forest had the lowest transpiration rates during the dry and heatwave periods. These differences were mainly due to tree species, their drought strategies (Gillner et al., 2017) and growing conditions of the sites, particularly soil moisture availability. Lower transpiration in *Betula pendula* at the Forest site indicates that *Betula*



*pendula* trees growing in an urban forest do not cool the environment as much as *Tilia* or *Malus*. Several previous studies have
reported that the transpirational cooling effect of urban trees during hot and dry days increases or sustains the transpiration
rates in order to prevent excessive heat accumulation (Gillner et al., 2015b; Duarte et al., 2016; Drake et al., 2018; Urban et al.,
2017; Ibsen et al., 2021). Keeping the stomata open in hot and dry conditions cools down the internal leaf temperatures enabling
maintaining photosynthesis (De Kauwe et al., 2019; Urban et al., 2017; Drake et al., 2018). The response depends on the species

tolerance to drought, water use efficiency, microclimatic conditions and site heterogeneity (Bussotti et al., 2014; Winbourne
et al., 2020; Rennenberg et al., 2006). Especially *Tilia cordata* is known for its anisohydric behavior (i.e. stomatal control is
not strong) during heat and drought and an associated increase in transpiration rates causing a cooling effect in different urban
conditions (Moser et al., 2017). However, also *Betula pendula*, which typically shows isohydric behavior (i.e. strong stomatal
control), increased $J_s$ during the heatwave and dry periods in the present study. Contrasting and species-specific response of

trees to heat and drought have been observed in several urban trees (Gillner et al., 2017; Osone et al., 2014), however here, a
rather constant pattern of heat and drought responses was observed between the studied species.

## 5   Conclusions

We conclude that the heat and drought that occurred in Helsinki  in 2021 were still not extreme enough to damage or dampen
the gas exchange functioning of urban trees. Against our hypotheses, photosynthetic potential did not reduce due to lowered

stomatal conductance during heatwave and drought conditions as the transpiration and photosynthetic potential during these
periods stayed high suggesting stable ecosystem services such as cooling and also carbon sequestration during rough condi-
tions. However, the significant role of VPD during heatwave and drought periods was well supported in our study but its overall
significance decreased during drought periods. The observed responses of tree transpiration and leaf gas exchanges during heat
and drought across the four urban green areas are mainly due to tree species type and VPD. Also, site-to-site variability in urban

tree water use was also largely explained by differences in tree size, varying growing conditions, and soil moisture availability,
apart from tree species type and VPD. As urbanization and the occurrence of climate extreme events are rising, particularly in
high-latitude regions, the role of urban green areas in mitigating climate change and cooling local microclimate is significant
in cities. Further studies of the cooling potential of urban trees will provide a better understanding and support the mitigation
and city planning in the future.

*Data availability.* Datasets of sap flux density, meteorological and leaf gas exchange measurements at the four urban sites in Helsinki is
stored in https://doi.org/10.5281/zenodo.7525319





## Appendix A

**Table A1.** The soil properties of the four urban vegetation sites. Soil sample analyzed from the top 30 cm of soil.

|  | Park | Street | Forest | Orchard |
|---|---|---|---|---|
| **Soil type** | Sand moraine | Fine sand moraine | Sand moraine | Sandy clay |
| **Bulk density (kg/l)** | 1.15 | 1.07 | 1.14 | 1.02 |
| **Main particle size distribution** | 66% sand 21% silt 8% clay | 48% sand, 26% silt, 11% clay | 71% sand 15% silt 11% clay | 27% sand, 31% silt, 42% clay |
| **Carbon content (%)** | 3.7 | 3.3 | 3.9 | 3.9 |
| **Nitrogen content (%)** | 0.252 | 0.168 | 0.329 | 0.32 |
| **C:N ratio** | 14.8 | 21.8 | 11.9 | 12.3 |
| **pH** | 5.6 | 7.2 | 6.5 | 5.9 |

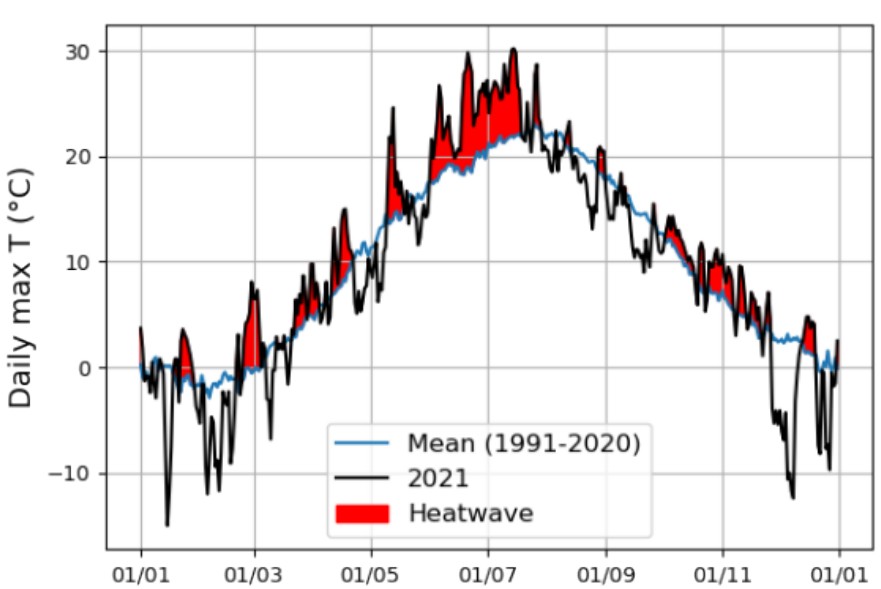

**Figure A1.** Heatwave detection using the daily maximum air temperature summer 2021 and the control period (1991-2020). The red color indicates the period where the daily maximum air temperature in the summer of 2021 exceeded the control period.


**Figure A2.** Meteorological condition at the Street site showing hourly a) air temperature (Air T), b) water vapor deficit (VPD), c) soil temperature (Soil T) and d) soil moisture measured and e) daily mean photosynthetically active radiation (PAR) and daily sum precipitation data measured at the SMEARIII station. The yellow markers in panel (c) denote the dates of manual leaf gas measurements.


**Figure A3.** Meteorological condition at the Forest site showing hourly a) air temperature (Air T), b) water vapor deficit (VPD), c) soil temperature (Soil T) and d) soil moisture measured and e) daily mean photosynthetically active radiation (PAR) and daily sum precipitation data measured at the SMEARIII station. The yellow markers in panel (c) denote the dates of manual leaf gas measurements.



**Figure A4.** Meteorological condition at the Orchard site showing hourly a) air temperature (Air T), b) water vapor deficit (VPD), c) soil temperature (Soil T) and d) soil moisture measured and e) daily mean photosynthetically active radiation (PAR) and daily sum precipitation data measured at the SMEARIII station. The yellow markers in panel (c) denote the dates of manual leaf gas measurements.





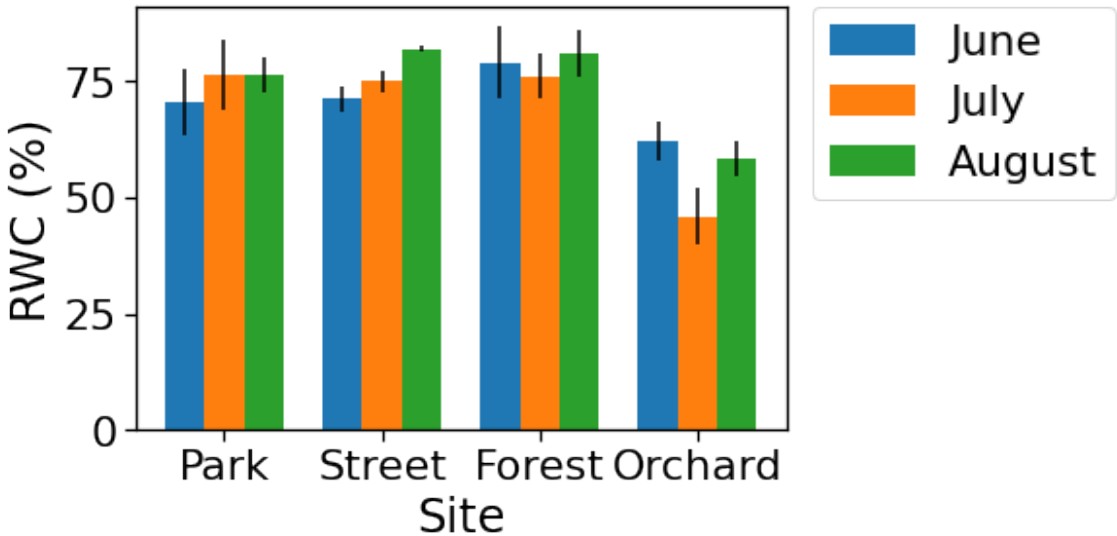

**Figure A5.** Monthly values of relative water content (RWC %) at the four urban sites.



*Author contributions.* Conceptualization: JA, LK, LJ; Data collection: JA, JS, YF, AK, EK, EV; Formal analysis: JA, JS, AK; Funding acquisition: LK, LJ; Supervision: LK, LJ; Visualization: JA, JS; Writing—original draft preparation: JA; Writing—review and editing: JA,
LK, LJ, YS, AL. All authors have read and agreed to the article.

*Competing interests.* The authors declare that there is no conflict of interest.

*Acknowledgements.* We thank the Academy of Finland (CarboCity project, decision numbers: 321527, 325549, 337549, and 337552), the Academy of Finland ACCC Flagship (decision numbers: 337549, 337552) and the Strategic Research Council working under the Academy of Finland (CO-CARBON project, decision numbers: 335201 and 335204). This project has also received funding from the European Union's
Horizon 2020 research and innovation programme under grant agreement 101037319 (PAUL project). We would like to thank Eki, Jarkko Mäntylä, Elisa Vainio and Teemu Paljakka for their technical support during the field measurements.



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
