# Peer review of "Sap flow and leaf gas exchange response to drought and heatwave in urban green spaces in a Nordic city"

_Biogeosciences, 2023_

## Referee Comment (RC2)

[referee-annotated manuscript omitted]

---

## Author Comment (AC1)

**RC1**: 'Comment on bg-2023-5', Anonymous Referee #1, 24 Feb 2023

We would like to thank the reviewer for the constructive and valuable comments and suggestions to improve the manuscript. Below we go through point-by-point our answers to the comments. The comment is always in black and our response in blue.

**General comments**

The introduction is not tailored to the research question but rather puts in various issues about urban trees, which are unfortunately sometimes wrongly cited (see also specific comments). The ways how cooling and shading might be affected by heat and drought (stomata, leaf senescence, hydraulic failure) are not properly described, nor are direct (heat and drought) and indirect impacts (nutrition, air pollution impacts) or impacts that may mitigate damaging influences (CO2, irrigation) properly differentiated.

We have improved the readability of the introduction towards the set research question. The last one of those, we revised to be more logical outcome of the introduction. Now it states,

L82: *"What are the main environmental drivers affecting transpiration rates during heatwave and drought in urban green areas?"*

We added information about tree cooling and shading role in urban trees and the tree responses to heat and dry with respect to the study research questions. We have also corrected and updated the references accordingly.

Also, hypotheses are not clearly specified and H1 and H3 seem to be the same anyway. Transpiration will be increased (vpd) or decreased (stomatal control), or both (in which cases)? Drought decreases stomatal conductance (old news) or photosynthetic capacity (and then impacts stomata)?

We have removed the original hypothesis H3 as it truly seemed to overlap with H1 as pointed out by the reviewer. We improved and specified only the H1 in the revised manuscript.

The methodology is problematic, trees were of different species at each site, so neither species differences nor site differences can be evaluated. Furthermore, trees varied considerable in height and age, and measurements were taken at different heights and also the stratification within the crown was not homogeneously done. In addition, some sites were irrigated while others were not.

We agree that the methodology had some limitations. Thus, we have removed the parts where there are unnecessary comparisons between sites and tree species in the study. Nevertheless, especially the trees in the park, the street and the forest are roughly equal in age (30-40 years) and soil type (sandy loam) and on the other hand, they represent typical species in their environment i.e., these specific green area types. For example, it is uncommon to have birch as a street tree whereas Tilia × europaea is clearly the most common street tree in Helsinki due to its capacity to bear compaction, salt, and pollution. At the same time, it is incredible to have an urban forest consisting of these non-native species in Helsinki. Therefore, we did not totally remove all text considering the site comparisons but revised the text throughout the manuscript in such a way, that is clear to the reader that these results arise not only from the site nor the species but a mixture of effects typical for them. In addition, we acknowledged that in the orchard, trees are older and the site. In any case, we focus mostly on the responses to heat and drought in the revised manuscript.

It is not clear, how the 'normalization' by vpd is actually done and if this is in accordance with theoretical considerations regarding the vpd response. In addition, it is not clear, how much drought stress was actually present at the different sites since only percentages of water are given without indication of how much water is available in absolute or relative (absolute in relation to total water holding capacity) terms.

We normalized the sap flux density using VPD by simply dividing the halfhourly $J_s$ by corresponding halfhourly VPD data. Please see our response to the comment below.

Overall, since vpd is not the only influence, which is necessary to consider in order to compare the different site and species responses, a more complex approach seems to be necessary in order to differentiate between site and species impacts. Perhaps, this means using a model that describes gas exchange based on climatic as well as soil conditions.

We agree that a more complex model describing both immediate and slow responses to environmental factors including phenological events such as growth stages of different organs could bring more insights into the observed effects. For example, the size of the reaction to a drought or a heatwave during the leaf-developing period most probably differs from the reaction seen later in the season. Such models could also be useful in an analysis differentiating the effect of site conditions and species, apart from VPD. However, including such modelling at this stage in this study is an enormous task requiring specific knowledge and skills. In this study, we included automatic sap flow data over four sites accompanied by manual leaf-level measurements of $CO_2$ and $H_2O$ gas exchange at different heights of the canopy, which we found to be a giant effort alone. Even though there are clear limitations in our study, such as missing process-based modelling, we find it valuable to share the unique data in a Nordic city experiencing extreme conditions and report that even

then, there were no clear indicators that trees in the different green spaces would notably decrease their ability to function and cool the environment. Therefore, as indicated already earlier, we decreased the focus on species-specific comparisons and physiological speculation and focused on the observed responses in transpiration and gas exchange. In addition, we included in the revised conclusions that there is a need for process-based modelling in order to validate the responses and omit the role of other driving environmental or e.g., phenological factors.

Despite the very different boundary conditions and species and the few samples for each condition, the discussion tries to differentiated between heat and drought effects although both influences were simultaneously occurring and can hardly be distinguished. This leads to counter-intuitive results (such as a restricting (significant?) role of soil moisture for forest gas exchange in the wet period) and very disputable conclusions (e.g. "severe weather events did not alter the stomatal action").

Thanks for the comment. We agree that the heat and drought are occurring simultaneously during a certain period. Although their effects can be hardly distinguished, we added new analysis and separate these stressful days into period of only heat, only dry and both heat-dry by the intersection of the heatwave days and dry period days which was derived as described in section 2.5. We analysed the response of sap flux density during only heat, only dry and both heat-dry periods at the four sites and found that the response during only heat was relatively higher than the other period at Park, Forest and Orchard; but at the Street site, there are no differences in the response of sap flux density during these periods (Appendix A6) and added the text accordingly in the manuscript (see below response)

The potential benefit of this study, which is the different response of species that a) do not close stomata, b) close stomata, c) are damaged by non-stomatal effects, could not be addressed due to the various degree of stress and different boundary conditions. Therefore, the conclusion is that species and site differences are causing the variation – which is probably true but could not really be shown given that neither species could be compared on different sites nor site influences could be evaluated using the same species. Accordingly, also the impacts that different species might have on their environment under heatwave conditions are highly speculative and are more based on literature than on measurements presented here.

We do acknowledge the worry about the solidness of our conclusions and therefore, we modified those according to the suggestion and removed the part where there is a comparison between sites and species. In addition, as stated already above, we focused there on the key message (just a minor decline if anything even though there were extreme conditions from the local perspective). We find that the revised discussion reflects the strength of this particular study without over-speculation and conclusions arising elsewhere.

**Specific comments (and some technical corrections)**

L1: I guess the role in offsetting CO2 emission is actually not important. If anything, consider pollution deposition effects. (see also L27)

Thank you for the comment! The offset by urban vegetation is found to be 6-14% of cities' anthopogenic emissions so even the value is low and they do not capture all emissions, we argue that their role is still significant (Hardiman et al. 2017, Vaccari et al. 2016, Havu et al. 2022). It is true that vegetation absorbs pollutants but at the same time, they emit BVOCs and particularly street trees can block the natural ventilation of street areas making the air quality worse. In any case, we added a comment amount the pollutant deposition to the sentence and removed the word important:

L1: *"Urban vegetation plays a role in offsetting urban CO2 emissions, mitigating heat through tree transpiration and shading, and acting as deposition surface for pollutants."*

L5: how many trees?

Three trees per site were selected. Information was now added to the abstract:

L5: *"We conducted sap flux density (Js) and leaf gas exchange measurements of three trees per species"*

L10ff: Do you say that street trees reduced their transpiration during a heatwave (relative to other sites) but not during dry periods? Can you actually differentiate between the two kinds of stress?

Thank you for the comment! We did not compare or differentiate the effect between heat and dry stress. In the results, we showed that there was no change or difference (same) in transpiration at the Street site during heatwave as compared to other heat periods, but the transpiration increased during dry period as compared to wet period.

However, as indicated in the general comments, we added new analysis to compare the effect of heat and dry separately by separating the period into three periods: Only heat, Only dry and both heat-dry by intersection of the heatwave and dry days which were defined earlier (as describe in section 2.5) and found that the effect of heat and dry periods did not change at the Street site but at the other three sites, the effect of heat stress on sap flux density was relatively higher than the effect of dry stress.

For clear understanding, we have reworded and added the sentences accordingly in the text. Also, an appendix figure was added as Appendix A6.

L13: *"but similar $J_s$ was observed at the Street site during the heatwave as compared to the non-heatwave period."*

L278: *"We separate the period into three periods: only heat, only dry and both heat-dry by intersection of the heatwave and dry days which were defined earlier (as describe in section 2.5) and analysed the effect of only heat, only dry and both heat-dry on sap flux density and found out that the effect of heat was relatively higher than the effect of dry and both heat and dry at Park, Forest and Orchard while at the Street site, the effect of heat, dry and both are similar (Appendix A6)."*

[Figure]

L15ff: Here you say that there is no effect on stomata during heatwaves - also for street trees. How does this fit to the indication of street trees having reduced their transpiration?

Thank you for this comment! There might have been a misunderstanding. In our result, we did not find any changes in the stomatal regulation during heatwave and dry period, thus indicating no significant effect on stomatal activities during these periods. Also, there was no change in sap flux density at the Street site during heatwave period. For more clarification, we have modified the sentence as shown in previous response.

L13: *"but similar $J_s$ was observed at the Street site during the heatwave as compared to the non-heatwave period."*

L17-20: In one sentence you say that drought limited transpiration in forests and in the following it is stated that drought was so mild that it could not affect stomata control. What did I miss here??

Thank you for the comment. Here in these lines, we want to describe the control of VPD over the sap flux density and at the Forest site, $J_s$ increases with the rise in VPD but saturates after certain level of VPD, and we think that it might be due to the low water availability at Forest site. We reworded the sentence as follow:

L19: "while at the Forest site, the increase of $J_s$ with the rise in VPD saturates after certain level of VPD, which might be evident due to low soil water availability at the Forest site during these hot and dry periods."

L28/29: regulating energy balance and cooling the surrounding is actually the same.

As suggested, energy balance is removed, and the sentence has been reworded as

L33: "they have the potential for carbon sequestration and storage, and regulation of water for cooling the surroundings"

L29: pollution deposition, not 'infiltration'

Infiltration changed to deposition, as suggested.

L32/33: Sorry, but none of the indicated references are saying that urban trees are important to mitigate the global GHG. In the contrary!

We are sorry for the confusion! We changed "mitigate the global GHG" to "cities' GHG emission" in the revised text. New references were updated and cited accordingly.

L34: *Urban trees also provide other ecosystem services such as cooling effect through shading, pollutant deposition and infiltration, aesthetics and recreation, buffer for noise and wind, and soil conservation (Brack, 2002; Jo, 2002; Jim and Chen, 2009; Pataki et al., 2009; Hardiman et al. 2017)."*

L35: human disturbances? Do you mean damages due to traffic or pruning by urban managers?

We have reworded the sentence to make it clear and added a reference (Czaja et al., 2020).

L45: *"In urban condition, trees are subjected to human disturbances such as construction activities, dense building, and vandalism (Czaja et al., 2020)"*

Czaja, M., Kołton, A., and Muras, P.: The Complex Issue of Urban Trees—Stress Factor Accumulation and Ecological Service Possibilities, Forests, 11, 932, https://doi.org/10.3390/f11090932, number: 9 Publisher: Multidisciplinary Digital Publishing Institute, 2020

L43-45: I cannot derive this conclusion from Bussotti et al.. Did I miss something?

Apologies for the misunderstanding. We have reworded the lines for better clarification.

L54: *"In urban areas, tree reactions to heatwaves are rarely studied, but trees in natural forests can adapt to the rising temperature by enhancing growth and utilized water more efficiently, provided enough moisture in the soil (Winbourne et al., 2020). The foliar temperatures of urban trees are often higher, which limits photosynthesis and transpirations through enzyme activity and stomatal regulation (Bussotti et al., 2014), hence such acclimation may not be apparent in urban settings."*

L65: What's the difference between research question 3 and 1/2? Do you want to address the impacts in combination?

Thank you for the comment. We would like to address the response or effect of heatwave and drought in urban trees transpiration and leaf gas exchange in research questions 1 and 2, while in research question 3, we would like to address the main environmental driver affecting the transpiration separately in heatwave and drought periods. For better clarification, we have reworded the 3$^{rd}$ research question as follow:

L82: *"What are the main environmental drivers affecting transpiration rates during heatwave and drought in urban green areas?"*

L60ff: Since the research capacity is limited, I think it is logical to assume that not all urban trees but a selection based of the most representative species is targeted. Please elaborate the text to make it more specific.

True, we are not measuring all trees. We modified the first sentence of that paragraph (original lines 59-60) where we state that we measured a set of typical urban trees as suggested.

L75: *"In this study, we measured the functions of a set of typical urban tree species, particularly their transpiration and leaf gas exchanges, during heatwave and drought periods in a boreal urban environment in Helsinki, Finland."*

L71-75: Shift to site description.

This line has been moved to site description as per reviewer's suggestion.

L104:*" The summer of 2021 was hot and dry as the mean air temperature in July 2021 was 21.6°C, being 21% higher than in July 2020 (16.7°C) and 19% higher than the average mean temperature in July (18.1 °C) during a climatic reference period (1991 to 2020). The total precipitation for the months of June and July 2021 (86 mm) was 51% lower than during June and July in 2020 (177 mm) and 27% lower than the average total precipitation in June and July during the climatic reference period (117 mm)."*

L80: sparse tree cover! Roadside single trees (not a plantation)!

Thank you for the correction. Roadside plantation has been changed to single line roadside trees accordingly.

L118: do you mean damages by pedestrians?

It has been reworded to "damage or disturbances by pedestrians" for more clarification.

L134: 'normalizing' flux density by dividing by vpd seems strange to me since velocity will likely have an S-shape response of vpd. Is this a common strategy (please cite) or may it be that you first would need to derive the dependency between both and then normalize by applying the respective function? Or did you do that as it seems later on that you fitted curves to this relationship?

Thank you for the comment! We did not normalize with any functions or with min-max normalization techniques. Here, we have only divided the half hourly $J_s$ data with their corresponding half hourly VPD in order to check the effect of VPD during these different periods and also to examine any dependency of $J_s$ on other environmental variables during these different periods. For better clarification, we have reworded the sentence as follows:

L157: "*In addition, we normalized the sap flux density using VPD by simply dividing the halfhourly $J_s$ by corresponding halfhourly VPD data. This is done inorder to assess the effect of VPD and also to examine the dependency of $J_s$ on other environmental variables during these heatwave and drought periods*"

L175ff: I don't understand this. What is the control period to define heatwaves? The long-term (30year) average for the respective season/month/specific days? The heatwave duration was more than one month? Can you provide explicit temperatures?

Thank you for your comment. Here, we have considered the control period as an average of daily maximum temperature for the last 30 years and added new sentence for more clarification and mean temperature values during the different heatwave periods have been provided accordingly.

L205: "*Here, we have considered the control period as an average of daily maximum temperature for the last 30 years. Accordingly, our study period was categorized into heatwave (21.9 °C; 17 June 2021 to 18 July 2021), pre-heatwave (16.8 °C; 1 June 2021 to 16 June 2021), post-heatwave (16.8 °C; 19 July 2021 to 31 August 2021) and no heatwave (17.5 °C; 1 July 2020 to 31 July 2020) periods.*"

L183: What do you mean with a 'further' separation of periods ... based on soil moisture? What is this give an absolute value that you indicate here – an arbitrary value between field capacity and wilting point? What is the water content relative to field capacity at this point?

Thank you for the comment. Here, we have removed the word "further" to avoid confusion. We meant to say that the period has been separated into dry and wet period based on no precipitation days and low threshold soil moisture data after considering the monthly SPEI value. Now, we have calculated the field capacity and wilting point based on soil texture type according to Hagemann & Stacke (2015) for the four sites and estimate the relative extractable soil water (REW) using soil moisture data, field capacity and wilting point. We considered REW threshold of 0.45 to determine the drought days along with the daily precipitation values. For better clarification, we added an estimation of wilting points at each site, and we reworded the sentences as follows:

L210: *"To determine the drought period, monthly Standardised Precipitation-Evapotranspiration Index (SPEI, Vicente-Serrano et al., 2010) was calculated and found out that June (SPEI = -0.7) and July (SPEI = -0.3) 2021 had mild drought conditions. According to the particle size distribution, all other sites were determined as sandy loams except for the Orchard which was clay. According to Hagemann, S Stacke T (2015), the wilting point and field capacity of sandy loam is 10 % and 22.9 %, respectively and of clay is 25 % and 38.4 %, respectively. We calculated the relative extractable soil water (REW) from soil moisture data, field capacity and wilting point of the site according to Granier et al., (1999). We considered days with precipitation less than 1 mm and mean REW in the depth of 10cm less than 0.45 as a dry period for all sites. As a result, the dry period was (22 June 2021 to 27 July 2021) and wet period (28 July 2021 to 31 August 2021)."*

L185: Are the hypotheses (if better defined) not better evaluated with transpiration than with sapflux density? Or by Water use efficiency?

Thank you for your comment which is correct! We have changed the transpiration into sap flux density in the hypothesis as per your suggestion.

L202: I would rather say site climate instead of microclimate (which is normally used for microsites such as canopy layers).

Thank you for the comment! We have changed microclimate to climatic conditions in the text as suggested.

L206: Soil moisture at what time of the year? Or do you mean maximum water content?

It is the average of soil moisture content during the summer months (June, July and August) of 2021. To make it clearer, below sentences have been added

*"L240 The mean of the meteorological variables (air temperature, VPD, soil temperature and soil moisture) during the summer months (June, July, August) of 2021 were considered for comparison."*

*"L246 Mean soil moisture content varied largely as the Orchard site had the highest (0.37 m3 m−3) and the Forest site the lowest (0.09 m3 m−3) whereas Park and Street sites had 0.13 and 0.22 $m^3 m^{−3}$ respectively."*

L211: Do you mean sapflow rate instead water use? (Water use would be expressed per m-2 ground area).

Yes true, it is sap flow rate. Thanks for the correction and now it has been changed to sap flow rate accordingly.

Figure 3: Strange that there is no increase in soil moisture at the 1$^{st}$ of July despite considerable rainfall. Any explanation for this?

Thank you for your comment! Actually, it does increase, particularly at the topsoil (10 cm depth) but at 30 cm depth soil, there is no sign of change as it seems the water didn't infiltrate well to the bottom soil and likely more of runoff or flash short rain on 1$^{st}$ July. I hope this explanation give more understanding on it.

Table 2: Do you mean average soil moisture within the indicated period. Please use average (relative) available soil moisture instead.

Thank you! We have changed the caption accordingly.

Table 2: *"Monthly mean air temperature (Air T, °C), mean vapor pressure deficit (VPD, kPa), mean soil temperature (Soil T, °C) and mean soil moisture content ($m^3 m^{−3}$) for the four study sites in 2021."*

L236: Still unclear how a normalization was done (see also comment to L134).

We only divided the half hourly $J_s$ data with their corresponding half hourly VPD in order to check the effect of VPD and also to examine any dependency of $J_s$ on other environmental variables during these different periods. Hope you will find this to be cleared now in the revised manuscript!

L282: vpd is due to impervious cover? Do you mean it is due to the increased temperatures that are caused by the impervious cover? Or are you assuming that vpd could be lower because of soil evaporation?

Thank you for the comment. Here, we observed that the VPD was higher at the Street site as compared to other site and we assumed that the high VPD was mainly due to larger impervious cover at the street site (fig 1c), which exacerbates the surface/ air temperature. For more clarification, the sentence has been reworded as follow:

L324: *"At the Street site, VPD was clearly higher as compared to the other three sites and this is likely due to the larger cover of impervious surface at the Street site where the air temperature is marginally increased"*

L289ff: Again, it is difficult to comprehend, what 'water use' means. If it is defined on the sapwood area, stem size indeed plays a major role (as indicated), but is this also the case if the water consumption/transpiration on a ground area basis would be calculated? I guess the latter is more important in the context of comparing site conditions.

Thanks for clarification! Here, we have changed the tree water use to sap flow rate or sap flux density accordingly, since we didn't really calculate with either sapwood area or crown area due to lack of these information and the literature derived sapwood area are not accurate to scale up to the tree level water use. We will not consider water use throughout the text to avoid confusion.

L291ff: Similarly, water availability is only meaningful if it is the water storage capacity minus the water bound by the wilting point. The definition is however, not clear.

Thank for the correction! True, water availability is meaningful in this context. However, we considered soil moisture content data due to lack of field capacity and wilting point data. Now, it has been changed to "soil moisture content".

L298ff: needs references. For Betula isohydry, I would recommend to consider Zapater et a. 2013. Tilia, however, seems also to be fairly isohydric (Leuschner et al. 2019), so I am not sure about the logic of the reasoning here.

Thank you for the suggestion. Now, we have added the suggested references accordingly in the text.

L301ff: What do you want to tell here? That Tilia may use more or less water than other trees? Or that site conditions might influence the transpiration rate of Tilia? But this is generally true for any species.

Thank you for the question! Here, we want to tell that in other studies in the street site of Munich and Helsinki, variability in tree wate use have been observed and are likely due to differences in tree species type. For better clarification, we have reworded the sentences as follows:

L351: *"Other studies in the streets of Munich and Helsinki have reported variability of transpiration rates, mainly due to the differences tree species. In Munich, the transpiration of Tilia cordata Mill. were three times higher than water use of Robinia pseudoacacia L. tree in the (Rahman et al., 2019) and in the street trees of Helsinki, Alnus glutinosa have four times higher tree water use than Tilia x vulgaris (Riikonen et al., 2016)"*

L323: replace 'cover' by 'represent' or similar (of course, one day does not cover the whole period)

As per the suggestion, 'cover' has been changed to 'represent' in the text.

L324ff: This is a conclusion that is contrary to the statement that Js/VPD declined during drought. Please provide an explanation for your conclusion. I can also not see where you see a similarity to Gillner et al. despite the very general fact that transpiration is mostly higher in summer. If this is about species comparison, the whole paragraph needs to be directed towards this.

Thank for this interesting comment! I think we have misinterpreted the results with the ratio between $J_s$ and VPD. The ratio value reduced during drought in all sites (fig 6b), which indicates the significant role of VPD during drought; however, while looking further into the relative importance of VPD in explaining the observed variation in sap flow during heatwave and dry conditions (Table 5), the influence of VPD varied at the four sites. We have also removed the reference Gillner et al 2015 as it is not relevant to the context.

For better clarification, we have reworded the sentences as follows:
L371: *"The ratio between $J_s$ and VPD was significantly reduced during all periods and at all sites, which indicates the substantial role of VPD; however, the relative importance of VPD over daily sap flow variation differed at the four sites and during different periods of heatwave and drought. The response of sap flow with VPD was less sensitive during heatwave and drought periods as compared to other heatwave periods and wet period (Table 4)."*

L333ff: You are probably indicating that the drought was not sufficient to cause damages to photosynthesis (non-stomatal effects, see e.g. Gourlez de la Motte et al. 2020), since you already discussed that a stomatal effect should have taken place. In addition, the stomatal control may be differently strong, depending on the isohydry or anisohydry of the species. The literature references should support your respective message and not only indicate similar results under possibly similar conditions.

Thank you for this interesting comment and the provided reference. We have added the below sentence with the suggested reference as per the suggestion.

L391: *"Interestingly, it was previously reported that the non-stomatal origin limitation was responsible for reductions of photosynthesis in temperate forest during European drought 2018 (Gourlez de la Motte et al. 2020) but these non-stomatal effects might play some role in our urban site but yet to study, and it may be also due to tree species behaviours (isohydryl or anisohydryl) towards drought response."*

L351: delete ', suggesting that ... at these sites' (redundant)

As suggested by reviewer, the words have been removed from the sentence accordingly.

L389-391: not shown in this paper

We have removed the sentences.

L391-393: this cannot be derived from this study

We have changed the sentences accordingly

L393/4: wishful thinking, not a conclusion

Thank you. We have changed it accordingly.

**Mentioned references**

Gourlez de la Motte L, Beauclaire Q, Heinesch B, Cuntz M, Foltýnová L, Sigut L, Manca G, Ballarin I, Vincke C, Roland M, et al. 2020. Non-stomatal processes reduce gross primary productivity in temperate forest ecosystems during severe edaphic drought. *Philosophical Transactions of The Royal Society B Biological Sciences* 375(1810): 20190527.

Leuschner C, Wedde P, Lübbe T. 2019. The relation between pressure–volume curve traits and stomatal regulation of water potential in five temperate broadleaf tree species. *Annals of Forest Science* 76(2): 60.

Zapater M, Bréda N, Bonal D, Pardonnet S, Granier A. 2013. Differential response to soil drought among co-occurring broad-leaved tree species growing in a 15- to 25-year-old mixed stand. *Annals of Forest Science* 70(1): 31-39.

Added references:

Hardiman, B. S., Wang, J. A., Hutyra, L. R., Gately, C. K., Getson, J. M., and Friedl, M. A.: Accounting for urban biogenic fluxes in regional carbon budgets, Science of The Total Environment, 592, 366–372, https://doi.org/10.1016/j.scitotenv.2017.03.028, 2017.

Havu, M., Kulmala, L., Kolari, P., Vesala, T., Riikonen, A., and Järvi, L.: Carbon sequestration potential of street tree plantings in Helsinki, Biogeosciences, 19, 2121–2143, https://doi.org/10.5194/bg-19-2121-2022, 2022.

Vaccari, F. P., Gioli, B., Toscano, P., and Perrone, C.: Carbon dioxide balance assessment of the city of Florence (Italy), and implications for urban planning, Landscape and Urban Planning, 120, 138–146, https://doi.org/10.1016/j.landurbplan.2013.08.004, 2013.

Brack, C. L.: Pollution mitigation and carbon sequestration by an urban forest, Environmental Pollution, 116, S195–S200, https://doi.org/10.1016/S0269-7491(01)00251-2, 2002.

Jim, C. Y. and Chen, W. Y.: Ecosystem services and valuation of urban forests in China, Cities, 26, 187–194, https://doi.org/10.1016/j.cities.2009.03.003, 2009.

Jo, H.: Impacts of urban greenspace on offsetting carbon emissions for middle Korea, Journal of Environmental Management, 64, 115–126, https://doi.org/10.1006/jema.2001.0491, 2002.

Pataki, D. E., Emmi, P. C., Forster, C. B., Mills, J. I., Pardyjak, E. R., Peterson, T. R., Thompson, J. D., and Dudley-Murphy, E.: An integrated approach to improving fossil fuel emissions scenarios with urban ecosystem studies, Ecological Complexity, 6, 1–14, https://doi.org/10.1016/j.ecocom.2008.09.003, 2009.

Granier A, Bréda N, Biron P, Villette S (1999) A lumped water balance model to evaluate duration and intensity of drought constraints in forest stands. Ecological Modelling 116: 269-283.

---

## Author Comment (AC2)

**RC2**: 'Comment on bg-2023-5', Laura Benegas, 10 May 2023

This paper addresses the issue of green infrastructure effect on ecosystem services like carbon sequestration, reports on the influence in water cycle processes like transpiration, and overall, provide insights on trees contributing to human well-being. The issue of urban trees improving such ecosystem services is fundamental to support the pertinence of nature-based solutions (NbS) Scientific evidence for these innovative approaches (NbS) are still scarce. The authors provide valuable evidence on this line of research.

I would like to thank the reviewer for the valuable comments and suggestions in the manuscript! We have agreed with the comments and corrected them accordingly.

Park and street sites have the same tree (species), but forest and orchard have two different tree species. This give a better condition to compare in a fair way the first two sites but could be questionable the comparison of the four sites together. In line 348 to 350, the authors state "The difference in saturation levels may be species-specific or caused by differences in the availability of soil moisture as the Orchard site had higher soil moisture than the Forest site during the heatwave and dry period". How can we separate these effects from the whole variables analyzed? Could the authors provide a section in discussion where it is addressed something like limitations and recommended improvements in this type of methodologies to cope with such result?

Thank you again for this comment! Yes, it is difficult to do the comparison of the four sites as the tree species are different with different site conditions. We have removed the comparison of site and species in the manuscript accordingly, as it is also not the main objective of the study. We have added a short paragraph about the limitation and a recommendation in the text as per the suggestion.

*L442: "In our study, we have observed these responses of urban tree transpiration and leaf gas exchanges pattern during heatwave and dry periods. However, there are challenges and limitations in the methods to conduct a detailed comparison of tree species. This is mainly because different tree species were measured at different sites. Also, the limited measurements of leaf gas exchanges could not be addressed more about water use efficiency during these local extreme periods. Further study with complex model capturing the effect of site conditions and tree species behaviour separately would be useful in addressing the main factor affecting the different responses of urban vegetation during the heatwave and dry period."*

Also, it will contribute to clarify the context if the authors add a line within the descriptive table 1, the main drought-resistant features of trees analyzed as their

defense strategies (escaping, avoiding or tolerating the loss of water) if there is something noticeable.

We have added the drought strategies behaviour of the studied tree species in the table caption, as per the suggestion.

There are differences in soil type between sites too, although all of them contains sand in some proportion, specifically speaking the authors are not comparing the exact soil conditions. The data provided on soil is reduced to the names of soil types. To provide more accurate information about the potential effects of soil type on the analysis, it would be desirable to also have a short description of the soil class (taxonomy) specially in terms of water retention properties.

Thank you for the comment! We agree that it would be informative with more soil properties parameters even though we are not comparing based on the soil conditions. Thus, we have added more information on soil properties with related to soil porosity, soil field capacity, wilting point and available water capacity of the four sites (together in Appendix A1) where we have calculated those parameters based on soil texture type and USDA classification scheme following Hagemann,S. & Stacke,T.(2015).

|  | Park | Street | Forest | Orchard |
|---|---|---|---|---|
| Soil porosity | 41.59 % | 41.59 % | 41.59 % | 46.13 % |
| Field capacity | 22.9 % | 22.9 % | 22.9 % | 38.4 % |
| Wilting point | 10 % | 10 % | 10 % | 25 % |
| Available water capacity | 12.9 % | 12.9 % | 12.9 % | 13.4 % |

*Available water capacity = Field capacity – Wilting point

In Figure 4, for the sake of clarity, it is important to explain here why there is no data the month 8 for street site at panel b. Figures should be interpreted as stand-alone piece in the paper, or self-explained.

Thank you for the suggestion! There were no data in August at Street site as the datalogger and sensor were broken. We have added this information in the figure caption as per the suggestion.

In table 3, please explain here why there is no pre-heatwave in street, and why is no data for no heatwave for orchard site. Such information also affects figure 5.

At the Orchard site, measurements started only in the summer 2021 and at the Street site, the intensive measurement days were started a bit late after the pre-heatwave period, thus no data for pre-heatwave at the Street site. Now, we have added this information in the Table 3 caption.

Added reference:

Hagemann,S., Stacke,T.(2015). Impact of the soil hydrology scheme on simulated soil moisture memory. Climate Dynamics. 44:1731–1750.https://doi.org/10.1007/s00382-014-2221-6

---

## Author Response (AR2)

Dear Authors

Congratulations, you are almost there!

I have now received the report of the one nominated referee on your first revision and it was suggested to accept for publication, after consideration of a number of points. The evaluation does not require new calculations and interpretations, but mainly tries to harmonise the presentation in the text with the actual results found. Mentioning of heat wave and drought responses has become less relevant after the clarifications made in discussion and the revision.

A number of additional suggestions on the presentation are made, which I ask you to consider or to clarify case by case.

With kind regards,
 Andreas

Dear Editor,

We would like to thank you for your report and support throughout the reviewing process. We have considered the points suggested by the reviewer in the revised manuscript accordingly. However, when it comes to using the heatwave and drought concepts, we pursue to use them also in the revised manuscript. We use standard and widely used methods for classifying the heatwave (definition by Fischer & Schar, 2010) and drought (SPEI values). Based on these, the local heatwave and drought were indeed experienced in the Helsinki region. The maximum temperature during the heatwave may not be as high as compared to southern regions but for Helsinki and its vegetation, the temperatures are clearly elevated reaching 30 °C and the daily temperatures are higher than 6 °C during the heat period when compared to the long term (1990 to 2020) reference data. Thus, the definition of Wikipedia for heatwave is also reached. In the revised manuscript, we clarified the definition of heatwave and emphasised the local nature of both heatwave and dry conditions. We added a new subsection in the revised manuscript for this and further motivate this in the detailed point-by-point responses below. The original comment is always in black and our response in blue.

With regards,

Joyson and co-authors

General comments

I acknowledge that more information about the boundary and driving conditions have been provided in the new version. In particular the results section has also much improved and looks mostly good now.

We would like to thank the reviewer for acknowledging the revised manuscript.

However, based on the knowledge provided and also consistent with the discussion as it is now, it seems necessary removing the indication of drought and heatwave from the manuscript throughout the text and concentrate on the potential of Finland's urban trees to provide the ecosystem service of heat mitigation under somewhat higher temperatures (such as expected in the future?).

As mentioned, we somewhat disagree with the suggestion to remove the indication of drought and heatwave. Our definition of heatwave is based on Fischer and Schär (2010) with over 1000 citations (in Google Scholar, see later). Moderate drought on the other hand is based on a commonly used drought indicator Standardized Precipitation Evapotranspiration Index (SPEI, Vicente-Serrano et al., 2010) calculated using local precipitation and potential evapotranspiration (PET). Thus, based on these two commonly used metrics we can quantify a local heatwave and drought. We have now sharpened the definitions in the manuscript (L205-220) and added a new section 2.5.

Also, we added text in the manuscript about the potential of urban trees in providing heat mitigation under higher temperatures in the Introduction and Discussion sections as per the suggestion.

L43-44: "Particularly, under extremely high temperatures, the potential of urban trees in heat mitigation has been shown to be significant (Gillner et al., 2015; Schwaab et al., 2021)"

L458-460: "The elevated levels of transpiration (represented by high $J_s$) observed in the studied green areas during the heatwave indicate the presence of transpirational cooling. This phenomenon could hold substantial promise for alleviating extreme heat conditions caused by exceedingly high temperatures."

Added reference:

Fischer, E. M. and Schär, C.: Consistent geographical patterns of changes in high-impact European heatwaves, Nature Geoscience, 3, 398–403, https://doi.org/10.1038/ngeo866, number: 6 Publisher: Nature Publishing Group, 2010.

Gillner, S., Vogt, J., Tharang, A., Dettmann, S., and Roloff, A.: Role of street trees in mitigating effects of heat and drought at highly sealed urban sites, Landscape and Urban Planning, 143, 33–42, https://doi.org/10.1016/j.landurbplan.2015.06.005, 2015

Schwaab, J., Meier, R., Mussetti, G., Seneviratne, S., Bürgi, C., and Davin, E. L.: The role of urban trees in reducing land surface temperatures in European cities, Nature Communications, 12, 6763, https://doi.org/10.1038/s41467-021-26768-w, number: 1 Publisher: Nature Publishing Group, 2021.

As it is, the manuscript states that heatwave and drought conditions were observed but impacts indicated that neither heat nor drought stress occurred – which is irritating at the least. I can corroborate my opinion by the authors definition of a heatwave as just 'exceeding the daily maximum air temperature of the control period'. The general definition of heatwave, however, is a 5 oC difference (see e.g. Wikipedia). The July 2021 seemed to be 3.5 oC warmer than the average from the 30 years before but the actual difference of the period defined as 'heatwave' (17.06-18.07.) is not given in the manuscript. If the weather doesn't meet the general definition, I strongly recommend to avoid the term 'heatwave' in this manuscript.

The 3.5 °C difference is based on the mean monthly temperature difference between July 2021 and July of the control period. When we look daily level, larger differences between the summer 2021 heatwave and the control period are seen (see below). We apologise for the misleading presentation. We have now added the actual temperature difference during the heatwave in the revised manuscript (L213).

We followed the Fischer and Schär (2010) definition of heatwave where heatwave is defined as the spell of at least six consecutive days with maximum temperatures exceeding the local 90th percentile of the control period. In our study, a heatwave was defined as a period of consecutive of at least 6 days when the local daily maximum air temperature of the year (2021) exceeds the daily maximum (100th percentile) air temperature of the control period (1991-2020).

Based on this definition, the daily maximum temperature for the period (17 June 2021 to 18 July 2021) exceeded the daily maximum air temperature of the control period (1991-2020) and stayed continuous at least for 6 days. The daily maximum air temperature during this heat period ranged from 20.5 to 30.2 °C, with a mean daily difference of 6 °C (ranging from 1.8 to 10.8 °C) with the control period. Accordingly, the period was categorized into heatwave (26.4 °C; 17 June 2021 to 18 July 2021), pre-heatwave (21.5 °C; 1 June 2021 to 16 June 2021), post-heatwave (20.4 °C; 19 July 2021 to 31 August 2021) and no heatwave (19.6 °C; 1 July 2020 to 31 July 2020) periods for comparison. The difference is also visualized in the figure below (Fig. C1).

[Figure]

*Figure C1: The daily maximum air temperature from 17 June to 18 July in 1991-2020 (black), in 2020 (red) and their difference (yellow bar). DOY indicates the Day of Year.*

Thus, looking at the daily level, also the Wikipedia definition for heatwave is met as the average difference of daily maximum temperature exceeds more than 5 °C the control period during the local heatwave period.

In the revised manuscript, we have separated the description of heatwave and drought into a new subsection for a clear definition of Heatwave detection.

*L230 -236: "According to Fischer and Schär (2010), a heatwave is defined as a spell of at least six consecutive days with maximum temperatures exceeding the local 90th percentile of the control period. Accordingly, in our study, heatwave (Appendix A1) as defined against a control period spanning from 1991 to 2020. Further, our study period was categorized into four periods: heatwave (17 June 2021 to 18 July 2021), pre-heatwave (1 June 2021 to 16 June 2021), post-heatwave (19 July 2021 to 31 August 2021) and no heatwave (1 July 2020 to 31 July 2020) periods with mean daily maximum air temperatures of 26.4 °C, 21.5 °C, 20.4 °C and 19.6 °C respectively. The daily maximum air temperature during the heat period ranged from 20.5 to 30.2 °C, with a mean daily difference of 6 °C (ranging from 1.8 to 10.8 °C) above the average temperature in the control period."*

I also recommend also to be careful with term 'drought'. The respective 2-month period is characterized to have 86mm rainfall (instead of usually 117). The estimated soil water availability hardly decreased below 45 % relative available water. Thus, the results

basically show that any increase in sap flux is due to a higher vpd resulting from higher temperatures.

Thank you for your comment! It is correct that the respective 2-month period experienced 86 mm rainfall during 2021 as compared to 117 mm for the climatic reference. We agree that the difference between these values is not dramatic. Monthly precipitation is, however, complicated indicator for drought as precipitation might not be evenly distributed throughout the period. For example, if the rain events take place at the beginning of the first month and late in the second month, there is a great change that the vegetation suffers from drought in between, especially if it takes place in the middle of the growing season with high evaporative demand. Here, Standardised Precipitation-Evapotranspiration Index (SPEI) calculated based on precipitation and potential evapotranspiration (Thornthwaite, 1948) indicated moderate drought during June and July 2021. We calculated the monthly SPEI based on the difference between daily precipitation and potential evapotranspiration (PET). PET was calculated using daily average temperature based on Thornthwaite (1948) equation. SPEI values in our study site fell into the moderate drought category, with SPEI = -1.4 in June and SPEI = -0.8 in July 2021.

We further used the relative extractable soil water threshold and no precipitation days to detect the days of dry or drought periods during these 2 months (June and July 2021).

We added the description of drought detection in the revised manuscript, in a similar manner as the heatwave (see the comment above).

*L237-243: "To determine the drought period, a monthly Standardised Precipitation-Evapotranspiration Index (SPEI, (Vicente-Serrano et al., 2010) was calculated, indicating that June (SPEI = -1.4) and July (SPEI = -0.8) 2021 had moderate drought conditions. Here, we considered days with precipitation less than 1 mm and mean relative extractable soil water (REW) at the depth of 10 cm less than 0.45 as a dry period for all sites. As a result, the dry period was from 22 June 2021 to 27 July 2021 and wet period from 28 July 2021 to 31 August 2021. We calculated the REW from the soil moisture data, field capacity and wilting point of the site according to Granier et al. (1999), where the wilting point and field capacity of sandy loam (Park, Street and Forest) are 10 % and 22.9 %, respectively and those of clay (Orchard) are 25 % and 38.4 %, respectively based on Hagemann and Stacke (2015)."*

The relatively small decrease of stomatal conductance in park and orchard trees is a bit irritating and might indicate a beginning soil water influence or a stronger isohydricity of Tilia cordata and Malus compared to the other two species. The impact, however, is obviously too weak to characterize this as drought stress.

We agree with the reviewer that the relatively small decrease in stomatal conductance in park and orchard might be due to isohydricity of the tree species of *Tilia cordata* and *Malus sp*. However, the influence of soil moisture during the dry period seemed to impact less the stomatal

conductance and thus no visible impact of the stress. We reworded the sentences in the revised manuscript.

*L420-422: "At the Park site, the relative decrease in gs during the dry period might be due to the influence of soil moisture reduction during the dry period and also due to the isohydric behaviour typical for Tilia cordata growing in the Park.".*

Specific comments:

According to the general comments, wording needs to change in all parts of the manuscript, and it is too tedious to make specific suggestions basically for each sentence. Therefore, I only indicate what I feel sounds particularly strange in every section.

Thank you for the comment. We have now carefully checked the language and improved it accordingly throughout the manuscript.

Abstract

- Imprecise problem definition (L4, 5)

We reworded the sentences as below:

*L2-4: "The frequent occurrence of heatwaves and concurrent drought conditions significantly disrupts the processes of urban trees, particularly their photosynthesis and transpiration rates. Despite the pivotal role of tree functioning in delivering essential ecosystem services, the precise nature of their response remains uncertain."*

- To detailed methodology (three trees)

We reworded accordingly

*L7-10: "We conducted sap flux density (Js) and leaf gas exchange measurements of four trees species (Tilia cordata, Tilia × europaea, Betula pendula, Malus spp.) located in different urban green areas (Park, Street, Forest, Orchard) in Helsinki, Finland. Measurements were made, over two contrasting summers 2020 and 2021."*

- Exaggerated boundary conditions (heatwave, drought)

We have added local heatwave and also please see the above comments for heatwave and drought description.

- It is irritating to see the results obtained for the street trees highlighted here.

Here, we have highlighted the results for all the sites and reworded the sentence for better clarification.

*L14-16: "When comparing the heatwave and non-heatwave periods, a 35-67% increase in Js was observed at the Park, Forest, and Orchard locations, while the Street site exhibited comparable values."*

Introduction

- I still think that the 'vital role in compensating urban CO2 emissions is exaggerated (e.g., Ariluoma et al. indicates 0.1 % for Helsinki residential greening, Street trees compensate for 0.08% of the transport sector in Bolzano). Don't mix up carbon stocks with net sequestration rates. It is also not important to dive on this because the current study only investigates the water balance.

The contribution from urban vegetation in offsetting anthropogenic emissions of cities ranges between 6-14% (Hardiman et al. 2017, Vaccari et al. 2016, Havu et al. 2022). Despite the modest magnitude and incomplete coverage of all emissions, we contend that their role remains noteworthy. We have reworded the sentence.

*L34-36: "Urban green spaces have a role in offsetting urban CO2 emissions and alleviating the urban heat island (UHI) effect, given their potential for carbon sequestration, storage, and the regulation of water to cool their surroundings (Lindén et al., 2016; Bowler et al., 2010, Hardiman et al. 2017, Havu et al. 2022)."*

 Added references:

Hardiman, B. S., Wang, J. A., Hutyra, L. R., Gately, C. K., Getson, J. M., and Friedl, M. A.: Accounting for urban biogenic fluxes in regional carbon budgets, Science of The Total Environment, 592, 366–372, https://doi.org/10.1016/j.scitotenv.2017.03.028, 2017.

Havu, M., Kulmala, L., Kolari, P., Vesala, T., Riikonen, A., and Järvi, L.: Carbon sequestration potential of street tree plantings in Helsinki, Biogeosciences, 19, 2121–2143, https://doi.org/10.5194/bg-19-2121-2022, 2022.

Vaccari, F. P., Gioli, B., Toscano, P., and Perrone, C.: Carbon dioxide balance assessment of the city of Florence (Italy), and implications for urban planning, Landscape and Urban Planning, 120, 138–146, https://doi.org/10.1016/j.landurbplan.2013.08.004, 2013.

- Too much emphasize on damages and extreme conditions that may cause uncertainty,

which is not addressed in this article. Concentrate on conditions that may be on the verge of limitation such as water supply (due to pervious soils) and surface overheating (due to already increased temperatures and less options for transpiration cooling).

We have removed and reworded accordingly

*L51-55: "In urban conditions, trees are subjected to harsh environmental conditions, such as elevated air temperature, lower air humidity, limited soil water and nutrient availability, compared with surrounding areas (Nielsen et al., 2007). Climate extremes such as heatwaves and drought affect the physiology of urban trees and thus also their potential to mitigate the effect of and to adapt to climate change. Hence, it is important to understand the response of the physiological processes regulating urban trees' functioning during extreme climate events."*

- Shortening and more precision is needed (e.g., L60-68). Also, gas exchange (including transpiration) is still not a function but a process (that might serve a function).

We have reworded the sentence as per the suggestions.

*L99-100: "In this study, we measured a set of typical processes regulating the functions of urban trees, specifically transpiration and leaf gas exchange, during local heatwave and drought in the boreal urban environment of Helsinki, Finland"*

- The setting is not suitable to address the research questions (see above).

We clarified in the revised text that the heatwave was local in the revised manuscript. Please see the above comments for heatwave and drought description.

- Stomatal closure decreases photosynthesis, not photosynthetic potential (or better: capacity). And sapflux isn't the driver but the result. If you want to hypothesize that heat and/or drought is decreasing photosynthetic capacity (that's possible too), stomatal closure would be a result of this (not the other way around).

Within a day, the light intensity determines the rate of photosynthesis. If the stomata are (partially) closed, the capacity (or potential) of a leaf to photosynthesize is lower than when

those are closed. In practise, by measuring the light response curve of photosynthesis, we can determine the capacity (or potential) of a leaf to photosynthesize on that day whereas the varying solar intensity determines the actual rate of photosynthesis. In the analysis, we have analysed e.g. the derived maximum rate of photosynthesis which indicates indeed the capacity than actual photosynthesis during that day as we did not want the possibly varying light conditions during the measurement day to cause extra variation in the results. Otherwise, we fully agree with the reviewer, and we have further clarified and reworded the hypothesis.

L110-112: *"H1) While increasing VPD during heatwaves increases the driving force for transpiration and thus sap flow, it also triggers stomatal closure, ultimately leading to a decrease in photosynthetic rates and a decoupling of VPD and leaf gas exchange rates"*

Methods

- Any references for indicating all sites/tree species as 'drought tolerant' in Table 1? Any definition for 'tolerant'? Wouldn't 'isohydric/anisohydric' be more indicative?

We have changed to isohydric or anisohydric in the Table 1 as per the suggestions and also added the references accordingly.

- Indicate that the values in Table 1 are means from three individuals (selected how?)

We have added in the table caption that the values are from mean of the three individuals where we conducted sap flux and leaf gas measurements.

- $CO_2$ concentrations for photosynthesis measurements were done under ambient conditions? Or set to 415ppm? If set – how? If not set – what were the ranges?

We have set the $CO_2$ concentrations at 415 PPM during the measurements as the instrument has the functionality for a user to adjust temperature, humidity, radiation and $CO_2$ concentration. Now, the words "ambient conditions" from the sentence is removed.

Results

- The climate in the two years is described two times in Methods as well as here. It is thus redundant.

We have removed it from the methods and reworded briefly:

*L110-111: " The summer of 2021 experienced elevated temperatures at 21.6°C, which were 21% higher, along with minimal rainfall at 86 mm, showing a 51% deficit compared to the reference period."*

*- I am still of the opinion that mean soil moisture is meaningless (Table 2). Since the soil texture has been estimated, it is possible to provide estimated relative or absolute available water instead. It seems that this has been provided in the appendix but should be in the main document.*

We have added the available water capacity in the main text.

*L269: "The soil water availability varied from 12.9% to 25% at the four sites (Appendix A1)."*

*- That vpd is not significant as a driver for street trees at some time seems purely related to data availability. The impression should be avoided that the data actually show that there is no relation between vpd and sapflux.*

We have reworded the sentence to avoid the misinformation.

*L298: "During wet period, VPD explained a higher share of the variation of $J_s$ at Park, Forest and Orchard site. At Street site, data availability was low and thus, the relationship could not be tested during wet period (Fig. 7)."*

*Discussion*
*- If any, a summary in the beginning of a discussion should be brief and should avoid conclusions (that come in the end). In any case, it should be clearly indicated that the hot (not the dry!) conditions drive water use (measured AS sapflux, or measured with sapflux METHODOLOGY).*

We have modified the summary according to the comment.

*L346-350: "In this study, we assessed the response of urban tree water use and leaf gas exchange during hot and dry summer 2021, across four urban green areas in Helsinki. Results indicated increased sap flux density in trees during the hot and dry periods, whereas carbon assimilation, stomatal conductance, and leaf-level transpiration remained largely unaffected. Sap flux density increase in such periods varied among the studied urban sites with distinct tree species and growth conditions. VPD emerged as the primary factor influencing tree water use in the heatwave and dry conditions. "*

*- I generally fully agree with the statements about the overall findings, but the word 'adaptations' is clearly not used in the correct frame. Observations just didn't indicate any stress apart from a mild stomata response.*

We have removed the word "adaptations" and reworded accordingly.

 L350-352: *"In summary, the urban trees in our study exhibited typical functioning during the summer of 2021, suggesting that the hot and dry conditions did not induce significant physiological changes or adjustments. However, we found some interesting insights that we will discuss further in this chapter."*

*- I appreciate highlighting the also investigated differences between the urban green sites with regard to climate. (But check the logic throughout the text. For example, it should be the temperature that is affected by pervious cover at the street site (also in the example of Los Angeles), which is then affecting vpd. If the temperature difference is really only marginal, a reason for the vpd effect is missing and needs to be provided).*

We agree with the reviewer and reword the sentence there to highlight the role of temperature as the driver for increased VPD and checked the logic throughout the text.

Line 370-371: *"At the Street site, VPD was higher than at the other three sites because of the higher air temperature likely due to the larger cover of impervious surface."*

*- Similar as has been stated before, the decrease of soil moisture is not a good indication of how much water is still there to transpire. Nevertheless, a decrease of 58 to 62 percent also give room for the assumption that there is quite some remaining water to transpire left.*

 We fully agree that relative water content or some other similar index describing the water available from plants and microbes considering also the soil water holding properties is better indicator that soil moisture content alone. Nevertheless, we think that there was a slight misunderstanding here as we are stating in the text that the soil moisture decreased by certain percentage during the heatwave and drought whereas we did not state here the pre- and post-event soil water contents. We improved the sentence to avoid further misunderstanding.

Line 416-419: *"During the dry period of this study, soil moisture content was notably reduced, ranging from 18% to 62% less compared to the wet period across all four locations. Similarly, during the heatwave period, the soil moisture content dropped significantly, ranging from 30% to 58% lower than the pre-heatwave period at all sites, except for the Street site."*

- Non-stomatal effects that were triggered before stomatal responses are possible but I don't see any indication of it. Does the An decreased relatively larger than gs? If not, it is probably not playing an important role here.

We agreed with the reviewer that there is no indication of non-stomatal effects. Thus, we have removed the part regarding the non-stomatal effects from the discussion section. Moreover, we also didn't see any changes in A/ci analysis.

- Note that the species-specific saturation of the sapflux/vpd curve can be explained by specific wood properties that allow for a certain maximum conductance only.

Thank you and fully agreed with the reviewer.

- Note that sapflux cannot be explained (generally) by many factors. You may use 'influenced' if necessary. On the other hand, this rarely seems relevant here.

We agree and here, we changed the word "explained" with "influenced" as .

- Some parts of the discussion can certainly be shortened, partly because of redundancy.

We have shortened the discussion part as per the suggestions.

Mentioned references

Ariluoma M, Ottelin J, Hautamäki R, Tuhkanen E-M, Mänttäri M. 2021. Carbon sequestration and storage potential of urban green in residential yards: A case study from Helsinki. Urban Forestry & Urban Greening 57: 126939.

Russo A, Escobedo FJ, Timilsina N, Zerbe S. 2015. Transportation carbon dioxide emission offsets by public urban trees: A case study in Bolzano, Italy. Urban Forestry & Urban Greening 14(2): 398-403.

---

## Author Response (AR3)

Dear Authors

thank you for the responses to the comments by referee 1 and the respective revisions.

Dear Editor,

We would like to thank you for your prompt reviewing process. We have considered the points suggested by you in the revised manuscript accordingly. We also found a few additional typing errors, which have now been fixed and visible with track changes. Also, figure #1 have been updated to allow readers with colour vision deficiencies correctly, as requested by the publisher.

Best regards,

Joyson & co-authors

I went quickly through the material and have the following comments:

in general consider speaking of "extreme weather" or "weather extremes" rather than climate extremes.

Thank you for the comment. We changed this throughout the manuscript.

15 - compared to your response letter you added "site which exhibited comparable values" after "while no significant change was seen at the Street" which is redundant.

We removed the line "which exhibited comparable values" as suggested.

70- "which further limits" - consider either "which may limit" or""which was shown to limit (REF)".

We changed this as suggested.

73- "in one of two ways:" can be omitted without loss (either or says exactly this)

We have omitted the words and changed the text as follows:

L73: *"A tree usually responds to drought either by avoiding a significant decrease in water potential and relative water content through stomatal closure at the cost of reduced photosynthesis"*

Tab. 1 and 474 : please avoid common language "fairly" and consider "mildly" as opposing to "strongly"

We changed this throughout the manuscript.

116- consider replacing "diverse" by "contrasting" or "characteristic"

We changed this to contrasting.

352 - if you agree, "uptake" would be a better term that "use"

We prefer to keep the term "use"

490 - keeping the long discussion about this in mind consider adding something like "This finding demonstrated that meteorological definitions of weather extremes must not be necessarily directly translate into extreme biological responses."
And maybe even - the Scandinavian perspective of what a heatwave can look like (rare T_air events at levels far lower than experienced in lower latitudes)

We fully agree with the editor's suggestion. Hence, we have the following lines in the conclusion section.

L423-425: *"Our finding demonstrated that meteorological definitions of weather extremes must not be necessarily directly translate into extreme biological responses and also the Nordic perspective on a heatwave is characterized by rare occurrences of air temperatures significantly lower than those seen in lower latitudes."*

With kind regards,
Andreas

Additional private note (visible to authors and reviewers only):
Hi Joyson and Co-authors,

although my comments are mainly non-technical, I have chosen "Publish subject to technical corrections". I did that to speed things up and I leave it to your own decisions, which of the comments you deem relevant to revise.
Please do not interpret my decisions in a way that I wasn't interested in a further discussion about these points with you.

Best wishes,
Andreas

Thank you again for your support and constructive suggestions.